# Perceptual and attentional impairments of conscious access involve distinct neural mechanisms despite equal task performance

Samuel Noorman[1,2]*, Timo Stein[1,2], Johannes Jacobus Fahrenfort[1,2,3]†, Simon van Gaal[1,2]†

[1]Department of Psychology, University of Amsterdam, Amsterdam, Netherlands; [2]Amsterdam Brain and Cognition, University of Amsterdam, Amsterdam, Netherlands; [3]Department of Applied and Experimental Psychology, Vrije Universiteit Amsterdam, Amsterdam, Netherlands

## eLife Assessment

This **important** study provides new insights into the mechanisms that underlie perceptual and attentional impairments of conscious access. The paper presents **convincing** evidence of a dissociation between the early stages of low-level perception, which are impermeable to perceptual or attentional impairments, and subsequent stages of visual integration which are susceptible to perceptual impairment but resilient to attentional manipulations. This study will be of interest to scientists working on visual perception and consciousness.

*For correspondence:
sgnoorman@gmail.com

†These authors contributed equally to this work

**Abstract** This study investigates failures in conscious access resulting from either weak sensory input (perceptual impairments) or unattended input (attentional impairments). Participants viewed a Kanizsa stimulus with or without an illusory triangle within a rapid serial visual presentation of distractor stimuli. We designed a novel Kanizsa stimulus that contained additional ancillary features of different complexity (local contrast and collinearity) that were independently manipulated. Perceptual performance on the Kanizsa stimulus (presence vs. absence of an illusion) was equated between the perceptual (masking) and attentional (attentional blink) manipulation to circumvent common confounds related to conditional differences in task performance. We trained and tested classifiers on electroencephalogram (EEG) data to reflect the processing of specific stimulus features, with increasing levels of complexity. We show that late stages of processing (~200–250 ms), reflecting the integration of complex stimulus features (collinearity, illusory triangle), were impaired by masking but spared by the attentional blink. In contrast, decoding of local contrast (the spatial arrangement of stimulus features) was observed early in time (~80 ms) and was left largely unaffected by either manipulation. These results replicate previous work showing that feedforward processing is largely preserved under both perceptual and attentional impairments. Crucially, however, under matched levels of performance, only attentional impairments left the processing of more complex visual features relatively intact, likely related to spared lateral and local feedback processes during inattention. These findings reveal distinct neural mechanisms associated with perceptual and attentional impairments and thus contribute to a comprehensive understanding of distinct neural stages leading to conscious access.

## Introduction

Conscious access to sensory input can be impaired in two distinct ways (*Dehaene et al., 2006*; *Lamme, 2010*; *Mashour et al., 2020*; *Northoff and Lamme, 2020*). Sensory input may lack sufficient bottom-up strength, or top-down attention may be directed elsewhere. Despite both cases resulting in a failure to perceive a stimulus, their underlying neural mechanisms are thought to be remarkably different. Influential theories of consciousness such as global neuronal workspace and recurrent processing theory propose four stages of neural information processing associated with distinct levels of bottom-up signal strength and top-down attention. These four stages can be investigated empirically by crossing 'perceptual' manipulations that degrade the strength of sensory input (e.g. reducing stimulus contrast, masking, continuous flash suppression) with 'attentional' manipulations that affect top-down attention (e.g. attentional blink, inattentional blindness, *Figure 1A*).

According to these theoretical models, all stimuli elicit feedforward information transfer from lower- to higher-level brain regions (*Figure 1A*, bottom row), but recurrent interactions are initiated only for stimuli with sufficient bottom-up strength (*Figure 1A*, top row). If stimuli are sufficiently strong and top-down attention is available, neural processing crosses a threshold, triggering a process termed global ignition, facilitating widespread recurrent interactions between frontal, parietal, and sensory cortices, yielding conscious access (*Figure 1A*, top left). Crucially, when top-down attention is lacking, frontoparietal network ignition is prevented, while local recurrent interactions within sensory brain regions remain relatively intact ('attentional blindness,' *Figure 1A*, top right) (*Dehaene et al., 2003*; *Marti et al., 2015*; *Sergent et al., 2005*; *Zivony and Lamy, 2022*). Weak stimuli result in the absence, or a substantial reduction, of local recurrent interactions ('perceptual blindness,' *Figure 1A*, bottom left) (*Fahrenfort et al., 2007*; *Joglekar et al., 2018*; *van Gaal et al., 2008*; *van Vugt et al., 2018*).

Although this framework is at the heart of influential theories of consciousness, the four stages of the model and their underlying neural mechanisms have rarely been investigated simultaneously within the same study (for an exception, see *Fahrenfort et al., 2017*). One challenge with comparing results across different studies, or even within a study, is that perceptual manipulations tend to impair overall task performance more than attentional manipulations, so that it may not be surprising to find that perceptual manipulations interrupt recurrent interactions to a greater extent than attentional manipulations. Given the right parameter settings, perceptual manipulations can be used to induce chance-level performance, while it is not possible to use attentional manipulations to drive behavioral performance down to chance, even when they are optimized fully. For this reason, attentional manipulations are often combined with post-hoc selection of a subset of 'blind' trials (e.g. attentional blink [AB]) or subjects (e.g. inattentional blindness) based on subjective awareness reports, a methodologically questionable practice that is susceptible to criterion confounds and introduces sampling bias that leads to underestimation of consciousness (*Fahrenfort et al., 2024*; *Peters and Lau, 2016*; *Schmidt, 2015*; *Shanks, 2017*).

Thus, when comparing perceptual to attentional manipulations, any (neural) effect could reflect differences in task performance rather than genuine differences between manipulations and hence stages in the model depicted in *Figure 1A*. While these issues with comparing conditions that differ in task performance in consciousness research have been acknowledged (*Lau, 2022*), they have rarely been addressed experimentally (*Kanai et al., 2010*; *Lau and Passingham, 2006*; *Meuwese et al., 2014*). For example, *Fahrenfort et al., 2017* found that illusory surfaces could be decoded from electroencephalogram (EEG) data during the AB but not during masking. This was taken as evidence that local recurrent interactions, supporting perceptual integration, were preserved during inattention but abolished by masking. However, masking had a much stronger behavioral effect than the AB, effectively reducing task performance to chance level. Indeed, a control experiment using weaker masking, which resulted in behavioral performance well above chance similar to the main experiment's AB condition, revealed some evidence for preserved local recurrent interactions also during masking. However, these conditions were tested in separate experiments with small samples, precluding a direct comparison of perceptual vs. attentional impairments at matched levels of behavioral performance. To test and further refine the four-stage model of consciousness in humans, the present study compared all stages within the same experimental setup, analyzing all trials and including all participants while carefully matching task performance between a perceptual manipulation (masking) and an attentional manipulation (attentional blink).

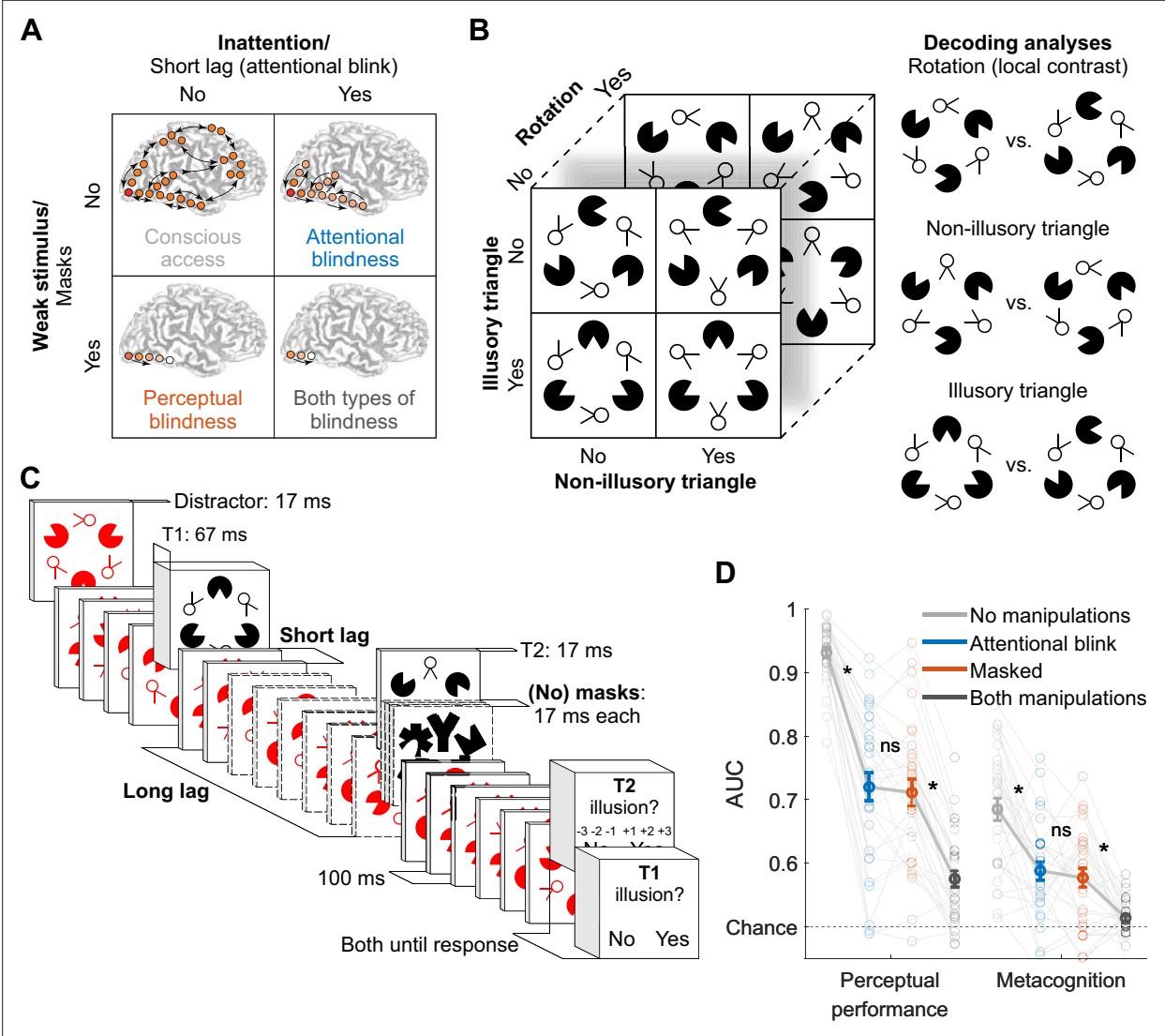

**Figure 1.** Experimental design and behavior. (**A**) Perceptual vs. attentional blindness in the four-stage model. A stimulus with low bottom-up strength (masked) is thought to interrupt local recurrent processing in sensory areas while leaving feedforward processing largely intact, while inattention (induced by the attentional blink) is thought to interrupt global recurrent processing between frontoparietal areas and sensory areas, while leaving local recurrent processing within sensory areas largely intact. Reprinted from *Dehaene et al., 2006* with permission from Elsevier. (**B**) Target stimulus set and decoding analyses. (**C**) Trial design. Note that the central fixation cross is not displayed for readability. (**D**) Perceptual performance refers to participants' ability to detect the Kanizsa illusion. Metacognition refers to participants' ability to evaluate their own performance using confidence judgments. Both perceptual performance and metacognition are measured as the area under the receiver operating characteristic curve (area under the curve, AUC). Error bars are mean ± standard error of the mean (n = 30). Individual data points are plotted using low contrast. Ns is not significant (p≥0.477, $BF_{01}$ ≥4.05). *p≤0.001.

The online version of this article includes the following figure supplement(s) for figure 1:

**Figure supplement 1.** Alternative measures of behavioral performance.

To reveal neural mechanisms associated with perceptual and attentional impairments, we combined a novel visual stimulus with features of different complexity with time-resolved decoding of these visual features from EEG data (*Figure 1B*). The target stimulus differed along three dimensions (illusory triangle, non-illusory triangle, and local contrast) that were independently manipulated. First, 'Pac-Man' stimuli could create either the perception of an illusory surface in the shape of a Kanizsa triangle when aligned, or not, when misaligned. Second, additional 'two-legged white circles' could

form either a non-illusory triangle when their line segments were aligned, or not, when the 'legs' were misaligned. Third, for the local contrast manipulation, the whole stimulus was rotated by 180 degrees, so that the same retinotopic positions had high contrast in one spatial configuration and low contrast when flipped 180 degrees.

Human neuroimaging and direct neural recordings in animals have shown that the Kanizsa illusion is supported by both lateral and feedback connections (*Halgren et al., 2003*; *Kok et al., 2016*; *Kok and de Lange, 2014*; *Lee and Nguyen, 2001*; *Pak et al., 2020*; *Wokke et al., 2013*), while the processing of collinear line elements (as in the non-illusory triangle) primarily relies on lateral connections (*Bosking et al., 1997*; *Gilbert and Wiesel, 1979*; *Li, 1998*; *Liang et al., 2017*; *Schmidt et al., 1997*; *Stettler et al., 2002*). Differences in local contrast are processed early in the visual system through feedforward connections and are resistant to masking (*Fahrenfort et al., 2007*; *Fahrenfort et al., 2017*; *Kandel et al., 2000*; *Lamme and Roelfsema, 2000*). Decoding the stimulus conditions of the illusory triangle, non-illusory triangle, and local contrast at different points in time, in combination with the associated topography, therefore, served as putative markers of these distinct neural processes (*Figure 1B*) and allowed us to test whether the effects of masking and the attentional blink followed the predictions of the four-stage model of consciousness.

## Results

### Masking and the attentional blink were matched in perceptual performance and metacognition

We recorded the EEG signal of 30 participants who identified the presence or absence of an illusory surface (triangle) in two black target stimuli (T1 and T2) that were presented amongst red distractors in a rapid serial visual presentation task (*Figure 1C*). We manipulated the visibility of T2 in two ways: masking the stimulus and manipulating attention, resulting in a 2×2 factorial design (*Figure 1A*). Specifically, T2 could be either masked or unmasked (perceptual manipulation), and T2 could be presented at either a long interval (900 ms) or a short interval (200 or 300 ms) after T1, inducing an attentional blink (AB) effect for the short T1-T2 intervals (*Raymond et al., 1992*). This design resulted in four conditions, which we from now on refer to as the masked condition (T2 masked at the long T1-T2 interval), AB condition (T2 unmasked at the short T1-T2 interval), no-manipulations condition (T2 unmasked at the long T1-T2 interval), and both-manipulations condition (T2 masked at the short T1-T2 interval). At the end of a trial, participants indicated whether each target (T1 and T2) contained an illusory surface or not. Importantly, mask contrast in the masked condition was adjusted using a staircasing procedure to match performance in the AB condition, ensuring comparable perceptual performance in the masked and the AB condition across participants (see Methods for more details).

Conscious access can be assessed not only based on perceptual performance but also through metacognitive sensitivity, the ability to evaluate one's own performance (; *Brown et al., 2019*; *Fleming and Lau, 2014*; *Lau and Passingham, 2006*; *Merikle et al., 2001*; *Seth et al., 2008*). Participants in our study provided confidence ratings on a 3-point scale (low, medium, high) for their responses to T2. To ensure that the distribution of confidence ratings in the performance-matched masked and AB condition was not influenced by participants anchoring their confidence ratings to the unmatched very easy and very difficult conditions (no and both manipulations condition, respectively), the masked and AB condition were presented in the same experimental block, while the other block type included the no- and both-manipulations condition.

We used area under the receiver operating characteristic (ROC) curve (AUC) as a shared metric for perceptual performance (detection of the Kanizsa illusion), metacognitive sensitivity, and EEG decoding (see Methods for details on the calculation of these measures). A repeated-measures (rm) ANOVA with the factors masking (present/absent) and T1-T2 lag (short/long) revealed, as expected, that both masking and the short T1-T2 lag impaired perceptual performance ($F_{1,29}=344.24$, $p<10^{-15}$, $BF_{10}=2.76 \times 10^{13}$ and $F_{1,29}=427.54$, $p<10^{-15}$, $BF_{10}=3.67 \times 10^{13}$) as well as metacognitive sensitivity ($F_{1,29}=50.78$, $p<10^{-7}$, $BF_{10}=1.18 \times 10^{5}$, and $F_{1,29}=47.83$, $p<10^{-6}$, $BF_{10}=3.28 \times 10^{4}$). Importantly, paired t-tests showed that we successfully matched the key conditions, the masked condition (masked, long lag) and the AB condition (unmasked, short lag) for perceptual performance ($t_{29}=0.62$, $p=0.537$, $BF_{01}=4.30$, *Figure 1D*, left) as well as for metacognitive sensitivity ($t_{29}=0.72$, $p=0.477$, $BF_{01}=4.05$; *Figure 1D*, right, see *Figure 1—figure supplement 1* for signal detection theory related measures of

performance). Thus, whereas in previous studies task performance was considerably higher during the AB than during masking (e.g. *Fahrenfort et al., 2017*), in the present study, the masked and the AB condition were matched in both measures of conscious access.

## Masking and the attentional blink leave local contrast decoding largely intact

To derive markers of the different neural processes from the EEG data, for each stimulus feature, we trained linear discriminant classifiers on the T1 data and tested them on the T2 data. Classifiers used preprocessed EEG activity (see Methods) across all electrodes. To leverage the similarities between T1 and T2 in task and stimulus context, all main analyses used T1 training data for T2 decoding. This approach minimized possible differences in conscious access and working memory demands between the training and test datasets.

For local contrast decoding, the classifier categorized stimuli as either pointing upwards or pointing downwards, thereby effectively decoding the stimuli's local differences in contrast at the top vs. bottom of the stimulus. Classification performance (AUC) over time was obtained, with peak decoding accuracy in a 75–95ms time window (*Figure 2A*, *Figure 2—figure supplement 1*, top). The peak in decoding accuracy was occipital in nature (see the covariance/class separability map of *Figure 2A*; *Haufe et al., 2014*), consistent with previous findings (*Fahrenfort et al., 2017*). We focused our analyses on the averages of this time window. An rm ANOVA with the factors masking (present/absent) and T1-T2 lag (short/long) revealed a weak but statistically significant effect of masking on local contrast decoding, with the Bayes factor indicating only 'anecdotal' evidence ($F_{1,29}=6.51$, p=0.016, $BF_{10}=1.31$), while the T1-T2 lag had no significant effect ($F_{1,29}=0.32$, p=0.578, $BF_{01}=3.45$). A paired t-test yielded no evidence for a difference between the performance matched conditions (masked vs. AB; $t_{29}=1.42$, p=0.166, $BF_{01}=2.08$; *Figure 2C*, 'Local contrast: 75–95 ms'). These results are in line with theoretical proposals and empirical findings that suggest limited effects of masking and attentional manipulations on perceptual processes that rely on feedforward connections (*Dehaene et al., 2006*; *Fahrenfort et al., 2007*; *Fahrenfort et al., 2017*; *Lamme, 2010*), contrasting with the robust effects of these consciousness manipulations on feedback processes reported in the following section.

## Stronger effect of masking than the attentional blink on early but not on late illusion decoding

Next, we trained a linear classifier on the T1 data to discriminate between the presence and absence of the Kanizsa illusion and tested it on each of the four conditions of the T2 data. The average of all four conditions revealed two prominent peaks in decoding accuracy, consistent with previous research (*Figure 2—figure supplement 1C*, top) (*Fahrenfort et al., 2017*). Based on this previous study, our analyses focused on the averages of the two-time windows that encompassed these two peaks: specifically, from 200 to 250 ms and from 375 to 475 ms after target stimulus onset. The covariance/class separability maps (*Figure 2B*) indicated that during the earlier time window (200–250 ms) classification mainly relied on occipital electrodes. Considering its timing, topology, and previous findings, this neural event likely reflects early sensory processes and may thus represent a marker for local recurrent processing (*Fahrenfort et al., 2017*; *Kok et al., 2016*; *Roelfsema, 2006*; *Wokke et al., 2013*; *Wyatte et al., 2014*). The timing and topology of the later neural event (375–475 ms) overlapped with the event-related potential component P300 (*Figure 2—figure supplement 3*), which is associated with conscious access (*Fahrenfort et al., 2017*; *Sergent et al., 2005*; *Weaver et al., 2019*) and may thus represent a marker for global recurrent processing.

We tested how the consciousness manipulations affected these putative markers of local and global recurrent processing (*Figure 2C*), again conducting rm ANOVAs with the factors masking (present/absent) and T1-T2 lag (short/long) that we followed up on with paired t-tests comparing the matched conditions (masked vs. AB). Importantly, we observed a distinct difference between the performance matched conditions in the first decoding peak, which was significantly impaired by both masking ($F_{1,29}=162.62$, $p<10^{-12}$, $BF_{10}=5.86 \times 10^{10}$) and T1-T2 lag ($F_{1,29}=78.07$, $p<10^{-8}$, $BF_{10}=6.03 \times 10^6$), but importantly, it was more affected by masking than by T1-T2 lag (interaction, $F_{1,29}=18.67$, p<0.001, $BF_{10}=149.92$) (*Figure 2C*, 'Illusory triangle: 200–250 ms'). Directly comparing the performance matched conditions, the first decoding peak was more strongly impaired by masking than by the AB ($t_{29}=4.66$, $p<10^{-4}$, $BF_{10}=3.81$). This suggests that the AB allowed for greater residual local

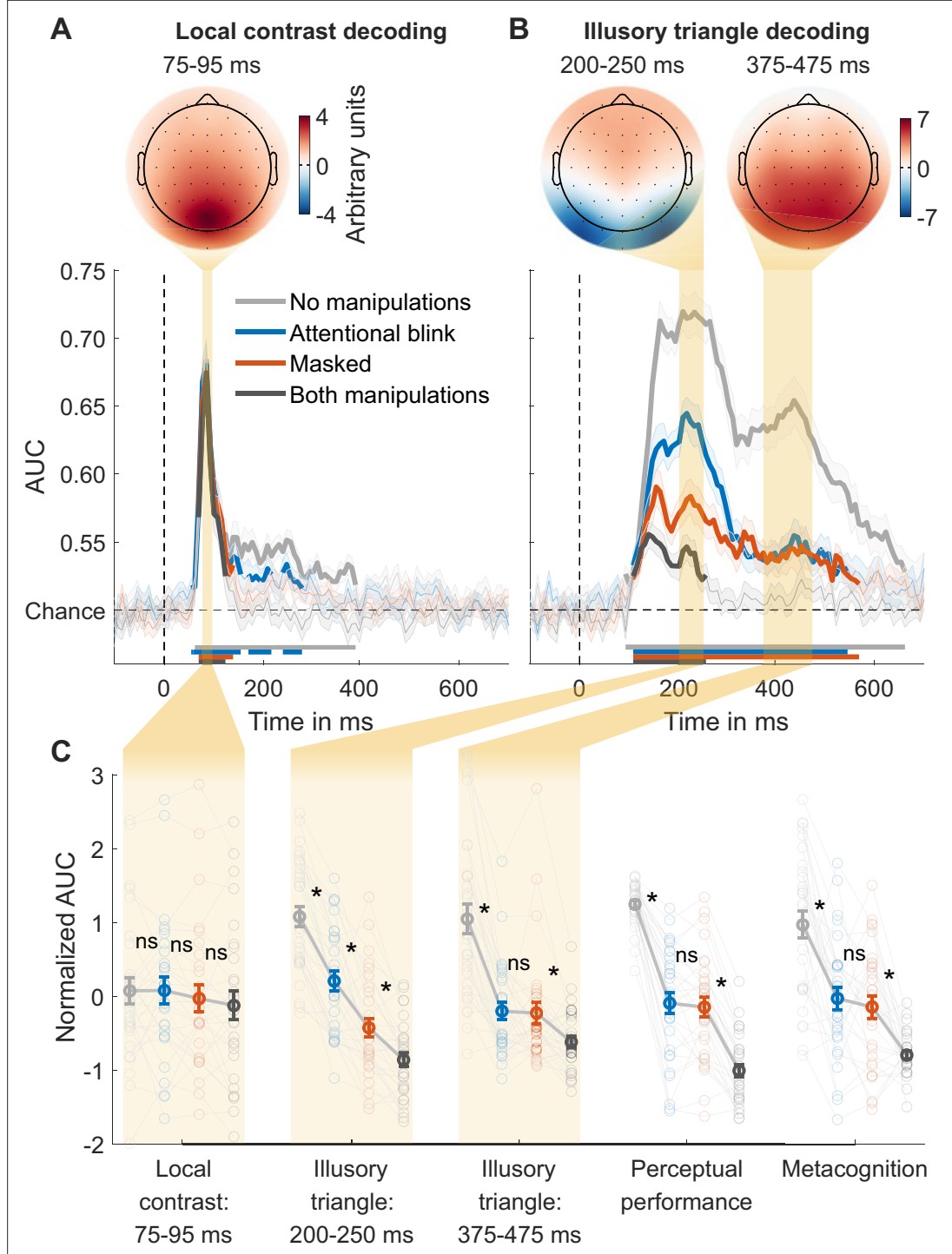

**Figure 2.** Local contrast and illusory triangle decoding using first targets as training data. (**A**) Local contrast decoding. (**B**) Illusory Kanizsa triangle decoding. For both features, covariance/class separability maps reflecting underlying neural sources are shown. Below these maps: mean decoding performance, area under the receiver operating characteristic curve (area under the curve, AUC), over time ± standard error of the mean (SEM; n = 30). Thick lines differ from chance: $p<0.05$, cluster-based permutation test. (**C**) Normalized (Z-scored) AUC for every measure: mean decoding time windows and two types of behavior. Each measure is Z-scored separately. Perceptual performance refers to participants' ability to detect the Kanizsa illusion. Metacognition refers to participants' ability to evaluate their own performance using confidence judgments. See *Figure 2—figure supplement 2* for the same analyses but then for off-diagonal decoding profiles. Error bars are mean ± SEM. Individual data points are plotted using low contrast. Ns is not significant ($p≥0.166$, $BF_{01} ≥2.07$). *$p≤0.002$.

The online version of this article includes the following figure supplement(s) for figure 2:

*Figure 2 continued*

**Figure supplement 1.** Averaged decoding accuracy peaks and time window selection.

**Figure supplement 2.** Local contrast and illusory triangle off-diagonal decoding using first targets as training data.

**Figure supplement 3.** Event-related potential component P300 derived from the first targets.

recurrent processing than masking, replicating the key finding by *Fahrenfort et al., 2017*. Importantly, the present result demonstrates that this effect reflects the difference between the perceptual vs. attentional manipulation rather than differences in behavior, as the masked condition and the AB condition were matched for perceptual performance and metacognition.

The pattern of results of the second peak was notably different. The second decoding peak was impaired by both masking ($F_{1,29}$=49.75, p<$10^{-7}$, $BF_{10}$=8.60 × $10^4$) and T1-T2 lag ($F_{1,29}$=78.48, p<$10^{-9}$, $BF_{10}$=4.32 × $10^5$), and the matched conditions (masked and AB condition) did not differ significantly from each other ($t_{29}$=0.21, p=0.837, $BF_{01}$=5.04) (*Figure 2C*, 'Illusory triangle: 375–475 ms'). Furthermore, another rm ANOVA comparing the first and second decoding peak between the matched conditions (masked/AB) revealed a significant interaction, reflecting a larger difference between the AB and the masked condition in the first than in the second decoding peak ($F_{1,29}$=31.53, p<$10^{-5}$, $BF_{10}$=1693.17). The pattern of behavioral results, both for perceptual performance and metacognitive sensitivity, closely resembled the second decoding peak: sensitivity in all three metrics dropped from the no-manipulations condition to the masked and AB conditions, while sensitivity did not differ significantly between these performance-matched conditions (*Figure 2C*). Two additional rm ANOVAs with the factors measure (behavior, second EEG decoding peak) and condition (no-manipulations, masked, AB) for perceptual performance and metacognitive sensitivity were performed. We excluded the both-manipulations condition from this analysis due to scale restrictions: in this condition, EEG decoding at the second peak was at chance, while behavioral performance was above chance, leaving more room for behavior to drop from the masked and AB condition. The ANOVAs revealed no significant interaction (performance: $F_{2,58}$=0.27, p=0.762, $BF_{01}$=8.47; metacognition: $F_{2,58}$=0.54, p=0.586, $BF_{01}$=6.04). This similarity between behavior and EEG decoding replicates the findings of *Fahrenfort et al., 2017* who also found a striking similarity between late Kanizsa decoding (at 406 ms) and behavioral Kanizsa detection. These results indicate that global recurrent processing at these later points in time reflected conscious access to the Kanizsa illusion.

Additional rm ANOVAs comparing the effect of the consciousness manipulations on local contrast decoding with their effects on the first and second illusion decoding peak showed that, compared to contrast decoding, both manipulations had stronger effects on the first illusion decoding peak (masking: $F_{1,29}$=99.35, p<$10^{-10}$, $BF_{10}$=1.73 × $10^8$; AB: $F_{1,29}$=38.95, p<$10^{-6}$, $BF_{10}$=1.43 × $10^4$) and on the second illusion decoding peak (masking: $F_{1,29}$=22.25, p<$10^{-4}$, $BF_{10}$=477.25; AB: $F_{1,29}$=49.60, p<$10^{-7}$, $BF_{10}$=3.59 × $10^4$). In the light of evidence showing that unconscious processing is susceptible to conscious top-down influences (*Kentridge et al., 2004*; *Kiefer and Brendel, 2006*; *Naccache et al., 2002*), we ran control analyses showing that early local contrast decoding was not improved by rendering contrast task-relevant (see **Appendix 1** and *Figure 3—figure supplement 1*), indicating that these differences between illusion and contrast decoding did not reflect differences in task-relevance. This suggests that masking and the AB specifically influenced local and global recurrent processing, respectively, while early feedforward processing was less affected.

While the findings presented so far used the T1 data to train the classifiers, we replicated all key findings when training the classifiers on an independent training set where individual stimuli were presented in isolation (*Figure 3A*, results in **Appendix 1** and *Figure 3—figure supplement 2*).

## Distinguishing collinearity and illusion-specific processing

The performance matched masked and AB conditions differed only in the first illusion decoding peak, our putative marker of local recurrent processing, which was markedly more impaired in the masked than in the AB condition. Next, we determined whether this effect reflected relatively preserved processing of collinearity or of the Kanizsa illusion during attentional impairment specifically. In our target stimulus, collinearity was present when the Pac-Man stimuli aligned, inducing the illusory Kanizsa triangle. Notably, collinearity was also present when the line segments of the 'two-legged white circles' of the stimulus aligned, forming the non-illusory triangle. Note that the line segments

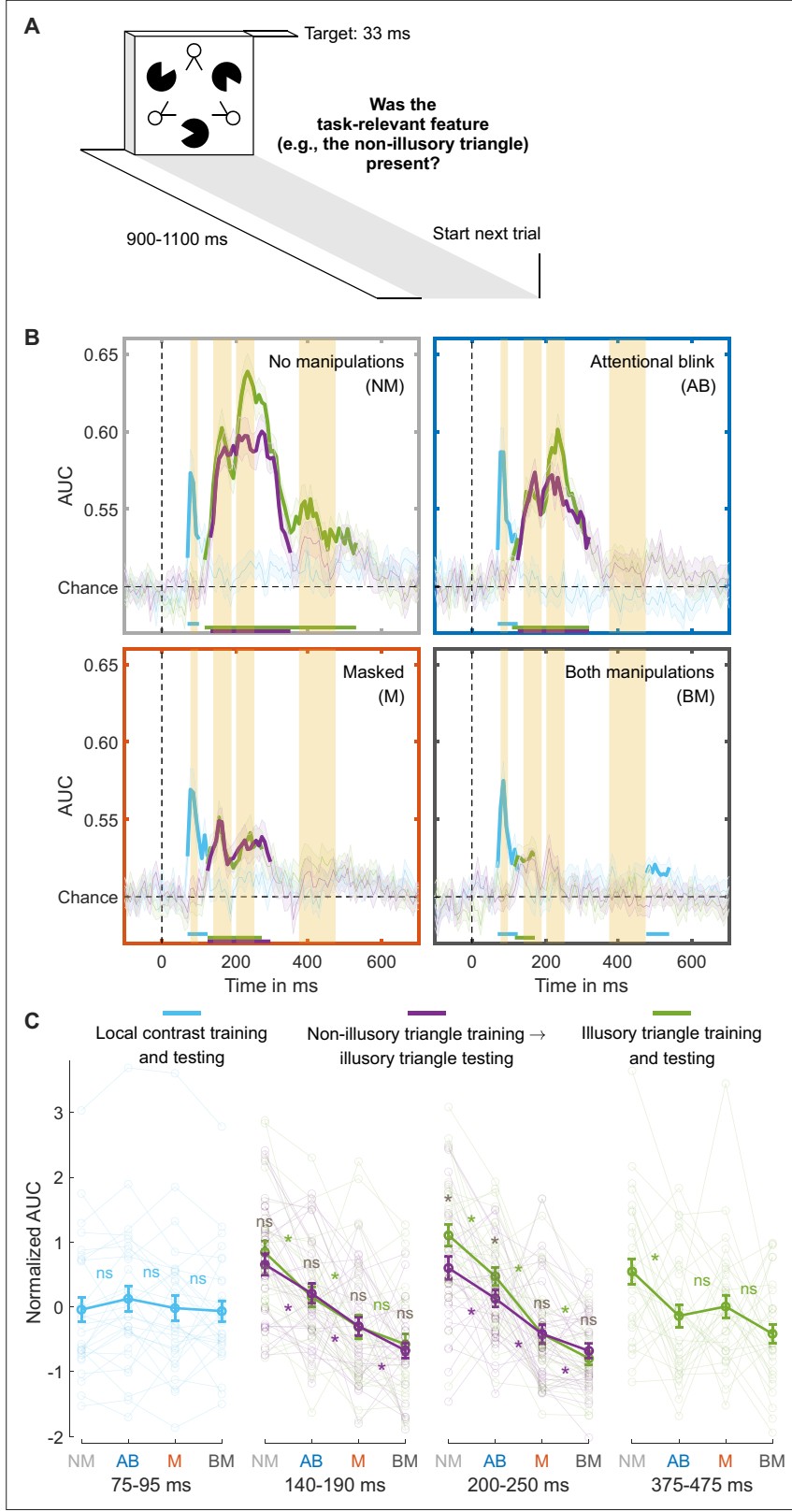

**Figure 3.** Separating collinearity and illusion-specific processes using the independent training dataset. (**A**) Trial design used for obtaining the independent training dataset. (**B**) Illusory triangle decoding, after training classifiers on the independent training set on either the non-illusory (collinearity-only, purple lines) or illusory triangle (collinearity-plus-illusion, green lines). For comparison, training and testing on local contrast is shown in light

*Figure 3 continued*

blue. Mean decoding performance, area under the receiver operating characteristic curve (area under the curve, AUC), over time ± standard error of the mean (SEM) is shown (n = 30). Thick lines differ from chance: p<0.05, cluster-based permutation test. The highlighted time windows are 75–95, 140–190, 200–250, and 375–475 ms, corresponding to separate panels in (**C**), which shows normalized (Z-scored) mean AUC for every time window. Each window is Z-scored separately. Error bars are mean ± SEM. Individual data points are plotted using low contrast. Ns is not significant (p≥0.084, $BF_{01}$ ≥1.26). *p≤0.048.

The online version of this article includes the following figure supplement(s) for figure 3:

**Figure supplement 1.** Local contrast decoding for each task-relevance condition from the independent training data and from another study.

**Figure supplement 2.** Local contrast and illusory triangle decoding using the independent training data.

**Figure supplement 3.** Control analyses for training on the non-illusory triangle using the independent training set and testing on the non-illusory triangle from the experimental session.

making up the triangle were equally long, and the spaces between them equally large, for the illusory and non-illusory triangles. The addition of this element of collinearity to our stimuli was a key difference to the study by *Fahrenfort et al., 2017*, allowing us to compare non-illusory triangle decoding to illusory triangle decoding in order to distinguish between collinearity and illusion-specific processing.

However, in the main RSVP task, the illusory triangle was task-relevant, while non-illusory triangles were always task-irrelevant. To equate the effect of task-relevance in the comparison, classifiers were trained on an independent training set in which each relevant stimulus feature was task-relevant. Specifically, in different experimental blocks, participants focused either on local contrast, the non-illusory triangle, or the illusory triangle (*Figure 3A*). We trained a classifier to distinguish between the presence and absence of the task-relevant non-illusory triangle (collinearity-only) and the same was done for the task-relevant illusory triangle (collinearity-plus-illusion). Then, both classifiers were used to decode the presence vs. absence of the illusory triangle in the main RSVP task (cross-task-decoding approach), which ensured that both training and testing were always performed on task-relevant stimuli. By comparing Kanizsa decoding performance in the RSVP task based on the collinearity-only classifier with decoding performance based on the collinearity-plus-illusion classifier, we effectively subtracted out the contribution of collinearity processing to illusion decoding, thereby isolating illusion-specific processing.

## Preserved collinearity and illusion-specific processing during the attentional blink

To determine a time window for (the start of) collinearity-only processing, we first trained and tested classifiers to distinguish present vs. absent non-illusory triangles (training and testing on non-illusory triangles only). We trained two classifiers, one on the T1 in the RSVP task and one on the independent training set (*Figure 2—figure supplement 1*), and tested their performance in decoding the non-illusory triangle in the T2 data, where this non-illusory triangle was always task-irrelevant. The results of both classifiers converged, and these analyses revealed a peak in decoding accuracy at ~164 ms (yielding a collinearity-only time window between 140–190 ms), right before the 200–250 ms time window of the first illusion decoding peak (*Figure 2—figure supplement 1B*). This 140–190 ms collinearity-only time window was also evident when classifiers were used to categorize the presence vs. absence of the Kanizsa illusion (*Figure 2—figure supplement 1C*, first time window), as well as in previous research (*Fahrenfort et al., 2017*), suggesting that collinearity processing also contributes to Kanizsa decoding.

We examined how decoding the presence vs. absence of the Kanizsa illusion in the RSVP task was affected by the consciousness manipulations, while training classifiers either on the illusory (collinearity-plus-illusion) or non-illusory (collinearity-only) triangle from the independent training set (as described above). *Figure 3B* shows the decoding accuracies of these analyses across the entire time window (purple and green lines). Follow-up analyses were performed using the 140–190ms collinearity-only window (see the previous paragraph). An rm ANOVA with the factors masking (present/absent), T1-T2 lag (short/long), and training set (illusory/non-illusory triangle) revealed that both masking and the short T1-T2 lag impaired decoding accuracy (masking: $F_{1,29}$=58.95, p<10$^{-7}$, $BF_{10}$=8.22 × 10$^5$; T1-T2

lag: $F_{1,29}$=24.90, p<10-4, $BF_{10}$=776.00). Furthermore, paired t-tests comparing the matched conditions (masked vs. AB condition) confirmed that decoding accuracy was more impaired in the masked than AB condition, both when training was done on the illusory ($t_{29}$=2.26, p=0.031, $BF_{01}$=0.58) and non-illusory triangle ($t_{29}$=2.78, p=0.009, $BF_{01}$=0.21; *Figure 3C*, '140–190 ms'). Focusing on the performance matched conditions and the role of the training set, an rm ANOVA with the factors condition (masked/AB) and training set (illusory/non-illusory triangle) on T2 illusory triangle decoding revealed no significant effect of training set ($F_{1,29}$=0.09, p=0.766, $BF_{01}$=4.22), i.e., no evidence for illusion-specific processing, and no significant interaction ($F_{1,29}$=0.04, p=0.837, $BF_{01}$=3.74). This demonstrates that neural processing in the 140–190 ms window indeed reflected collinearity-only rather than illusion-specific processing.

The marker for illusion-specific processing emerged later, namely in the 200–250 ms time window that encompassed the first illusion decoding peak reported above. As can be seen in *Figure 3B*, when no consciousness manipulations were applied (unmasked, long T1-T2 lag), there was significant illusion-specific processing, i.e., T2 illusory triangle decoding was better after training a classifier on the Kanizsa illusion (collinearity-plus-illusion, green line) than after training a classifier on the non-illusory triangle (collinearity-only, purple line) ($t_{29}$=4.22, p<0.001, $BF_{10}$=128.47). Turning to the performance matched conditions, an rm ANOVA with the factors condition (masked/AB) and training set (illusory/non-illusory triangle) on T2 illusory triangle decoding yielded a significant effect of condition ($F_{1,29}$=16.59, p<0.001, $BF_{10}$=90.73), with overall better decoding for the AB than for the masked condition, and importantly, a significant interaction ($F_{1,29}$=4.65, p=0.039, $BF_{10}$=2.19). *Figure 3B* shows that decoding after training on illusory triangles (collinearity-plus-illusion) was better than after training on non-illusory triangles (collinearity-only) for the AB ($t_{29}$=2.51, p=0.018, $BF_{10}$=2.78, *Figure 3B*, top right) but not for the masked condition ($t_{29}$=−0.02, p=0.982, $BF_{01}$=5.14, *Figure 3B*, bottom left). Thus, while illusion-specific processing was evident in the AB condition, it was fully abolished in the masked condition. Illusion-specific processing was not even affected by the AB, as an rm ANOVA with the factors condition (no manipulations/AB) and training set (illusory/non-illusory triangle) revealed no significant interaction ($F_{1,29}$=1.33, p=0.259, $BF_{01}$=2.30). The classifier trained on non-illusory triangles (collinearity-only) also performed better during the AB than during the masked condition ($t_{29}$=2.85, p=0.008, $BF_{10}$=5.43 *Figure 3C*, '200–250 ms,' purple line), hence both collinearity-only and illusion-specific processing were most strongly impaired by masking. Control analyses presented in **Appendix 1** and *Figure 3—figure supplement 3* demonstrate that cross-feature decoding can indeed isolate illusion-specific processes and does not reflect other, e.g., task- or attention-related, processes (for further control analyses testing the effect of task-relevance on local contrast processing, see **Appendix 1** and *Figure 3—figure supplement 1*).

Finally, we focused on the late 375–475 ms window, encompassing the second illusion decoding peak, which was directly linked to behavioral performance (see above, *Figure 3B*, last time window). Similarly, as above, illusory triangle decoding was now based on training the decoder on the illusory triangles from the independent training set. Replicating our main analysis, classifier performance was impaired by both masking ($F_{1,29}$=6.01, p=0.020, $BF_{10}$=2.37) and T1-T2 lag ($F_{1,29}$=10.10, p=0.004, $BF_{10}$=10.12), with no significant differences between the two performance matched conditions ($t_{29}$=−0.63, p=0.531, $BF_{01}$=4.27) (*Figure 3C*, '375–475 ms').

## Discussion

We demonstrate that perceptual and attentional manipulations, despite similarly impairing conscious access, exhibit distinct neural profiles in the brain. To investigate this difference, we decoded different visual features targeting distinct stages of visual processing from human EEG activity, while carefully matching a masked condition and an attentional blink (AB) condition in perceptual and meta-cognitive performance. Therefore, any observed neural difference between the masked and the AB condition could be unequivocally attributed to differences between attentional and perceptual manipulations of conscious access. While decoding of local contrast was barely affected by the two consciousness manipulations, early (200–250 ms, occipital) decoding of the illusory Kanizsa triangle was markedly more impaired in the masked than in the AB condition (*Fahrenfort et al., 2017*), even though behavior was matched. By contrast, later (375–475 ms, centroparietal) illusion decoding was similarly impaired by masking and by the AB, closely resembling their matched effects on behavior. Furthermore, we differentiated between collinearity-only and illusion-specific processing and found

that both processes were more strongly impaired by masking than by the AB. Notably, illusion-specific processing was unaffected by the AB, but completely abolished by masking.

Decoding of these different stimulus features at different points in time, together with their topography, may be regarded as markers of distinct neural processes. Based on neurophysiology and previous neuroimaging studies, the early decoding peak for local contrast decoding (75–95 ms after target stimulus onset) likely reflects feedforward processing (*Fahrenfort et al., 2007*; *Fahrenfort et al., 2017*; *Kandel et al., 2000*; *Lamme and Roelfsema, 2000*). The first illusion decoding peak (200–250 ms) may be regarded as a marker of local recurrent processing, while the second illusion decoding peak (375–475 ms) likely reflects global recurrent processing (*Fahrenfort et al., 2017*). Thus, our findings suggest that both perceptual and attentional impairments leave feedforward processing largely intact and similarly impair global recurrent processing, while local recurrent processing is markedly more impaired by perceptual than by attentional impairments.

This pattern of results is consistent with a previous study that used EEG to decode Kanizsa-like illusory surfaces during masking and the AB (*Fahrenfort et al., 2017*). However, the present study also revealed some effects where *Fahrenfort et al., 2017* failed to obtain statistical significance, likely reflecting the present study's considerably larger sample size and greater statistical power. For example, in the present study the marker for feedforward processing was weakly but significantly impaired by masking, and the marker for local recurrency was significantly impaired not only by masking but also by the AB, although to a lesser extent. Most importantly, however, we replicated the key findings that local recurrent processing was more strongly impaired by masking than by the AB, and that global recurrent processing was similarly impaired by masking and the AB and closely linked to task performance, reflecting conscious access. Crucially, having matched the key conditions behaviorally, the present finding of greater local recurrency during the AB can now unequivocally be attributed to the attentional vs. perceptual manipulation of consciousness.

Furthermore, the addition of line segments forming a non-illusory triangle to the stimulus employed in the present study allowed us to distinguish between collinearity and illusion-specific processing. Although the present EEG decoding measures cannot provide direct evidence for feedback vs. lateral processes, based on neurophysiological evidence that collinearity processing primarily relies on lateral connections (*Bosking et al., 1997*; *Gilbert and Wiesel, 1979*; *Li, 1998*; *Liang et al., 2017*; *Schmidt et al., 1997*; *Stettler et al., 2002*), while processing of the Kanizsa illusion involves both lateral and feedback connections (*Halgren et al., 2003*; *Kok et al., 2016*; *Kok and de Lange, 2014*; *Lee and Nguyen, 2001*; *Pak et al., 2020*; *Wokke et al., 2013*), the comparison of collinearity-only to illusion-specific processing may provide insight into the components of local recurrent processing: lateral and feedback connections (*Lamme et al., 1998*; *Roelfsema, 2006*). It should be noted that not all neurophysiological evidence unequivocally links processing of collinearity and of the Kanizsa illusion to lateral and feedback processing, respectively (*Angelucci et al., 2002*; *Bair et al., 2003*; *Chen et al., 2014*), so that overlap in decoding the illusory and non-illusory triangle may reflect other mechanisms, for example, feedback processes representing the triangular shapes as well. However, following our reasoning, the decoding results suggest that lateral processing occurred earlier (140–190 ms after target stimulus onset) than illusion-specific feedback processing (200–250 ms), in line with animal research (*Angelucci and Bressloff, 2006*; *Lamme et al., 1998*; *Roelfsema, 2006*).

Both lateral and feedback processing were more strongly affected by masking than by the AB, indicating that the 'attentional blindness' stage of the four-stage model of consciousness (*Figure 1A*) involves both lateral and feedback connections. Furthermore, masking had a stronger effect on illusion-specific feedback processing than on lateral processing. Notably, illusion-specific feedback processes were unaffected by the AB, but completely abolished by masking. As current theories do not distinguish between the roles of lateral vs. feedback connections for consciousness, the present findings may enrich empirical and theoretical work on perceptual vs. attentional mechanisms of consciousness (*Block, 2005*; *Dehaene et al., 2006*; *Hatamimajoumerd et al., 2022*; *Lamme, 2010*; *Pitts et al., 2018*; *Sergent and Dehaene, 2004*), clearly distinguishing the neural profiles of each processing stage of the influential four-stage model of conscious experience (*Figure 1A*). Along with the distinct temporal and spatial EEG decoding patterns associated with lateral and feedback processing, our findings suggest a processing sequence from feedforward processing to local recurrent interactions encompassing lateral-to-feedback connections, ultimately leading to global recurrency and conscious report.

Our results suggest that, compared to masking, the AB left local recurrent processing intact, while feedforward processing did not differ between the two manipulations. Local recurrent processing plays a critical role in perceptual integration, facilitating the organization of fragmented sensory information, such as lines, surfaces, and objects, into a coherent whole (*Roelfsema, 2023*). Our EEG decoding results support this notion, demonstrating that the AB allows for greater processing of collinearity and the illusion specifically within a time window spanning 140–250 ms after stimulus onset, likely reflecting sparing of local recurrent processes in visual cortex (*Fahrenfort et al., 2017*; *Kok et al., 2016*). This aligns with established models of the AB phenomenon, in which the AB reflects a late post-perceptual central bottleneck characterized by limited attentional capacity (*Shapiro et al., 1997*), so that sensory information presented during the AB can nevertheless undergo extensive processing, allowing for perceptual integration, possibly even leading up to semantic analysis (*Luck et al., 1996*). Clearly, these findings do not imply that unconscious high-level (e.g. semantic) processing can only occur during inattention, nor do they necessarily generalize to other forms of inattention. Indeed, while the AB represents a prime example of late attentional filtering, other ways of inducing inattention or distraction (e.g. by manipulating spatial attention) may filter information earlier in the processing hierarchy (e.g. *Luck and Hillyard, 1994* vs. *Vogel et al., 1998*).

Preserved local recurrent processing during the AB is also consistent with classic load theory (*Lavie and Dalton, 2014*), where increasing perceptual load (*Lavie and De Fockert, 2003*) more strongly reduces distractor processing than increasing cognitive load (e.g. by engaging working memory, as in our AB condition). According to this theory, perceptual and attentional manipulations serve as early and late filters for incoming sensory information, respectively, resulting in more extensive processing under inattention. Indeed, one of the few neuroimaging studies that included both manipulations found that only perceptual but not cognitive (working memory) load decreased fMRI activity in the parahippocampal place area in response to distractor scenes (*Yi et al., 2004*), although not all neuroimaging evidence is consistent with a stronger effect of perceptual than cognitive load (*Brockhoff et al., 2022*).

Furthermore, previous research has shown that the impact of inattention vs. masking can depend on the neural architecture required for the task at hand. For example, processes related to the detection of conflicting response tendencies, a hallmark of cognitive control and strongly associated with the prefrontal cortex (*Ridderinkhof et al., 2004*), are more susceptible to inattention, which reduces the depth of stimulus processing (*Nuiten et al., 2021*) than to masking, restricting recurrent interactions, but allowing for deep feedforward processing (all the way up to prefrontal cortex) (*Jiang et al., 2018*; *van Gaal et al., 2008*). Such deep feedforward processing can be sufficient for unconscious high-level processing, as indicated by a rich literature demonstrating high-level (e.g. semantic) processing during masking (*Kouider and Dehaene, 2007*; *Van den Bussche et al., 2009*; *van Gaal and Lamme, 2012*). Thus, rather than enabling deep unconscious processing, preserved local recurrency during inattention may afford other processing advantages linked to its proposed role in perceptual integration (*Lamme, 2020*), such as integration of stimulus elements over space or time.

Having delineated these distinct stages of feedforward, lateral, feedback, and global recurrent processing, one important avenue for future research is to distinguish between unconscious and conscious perceptual processes at these stages. Having opted to equate performance across manipulations in our study, behavioral performance was above the chance level for both consciousness manipulations. Follow-up research investigating perceptual integration of fully unconscious stimuli could address ongoing debates between influential theories of consciousness (*Ferrante et al., 2023*; *Mudrik et al., 2014*). The global neuronal workspace theory suggests a durable, yet unconscious processing stage (referred to as 'preconscious'), where the input is not globally available, and amplification through top-down attention is required for conscious access and report (*Dehaene et al., 2006*). In contrast, others have argued that already local recurrent interactions reflect subjective phenomenal experience (*Block, 2005*; *Lamme, 2010*). Moreover, markers like the P300 and ours for global recurrent processing may reflect functions not directly related to conscious experience, like report or decision-making (*Alilović et al., 2023*; *Canales-Johnson et al., 2023*; *Pitts et al., 2018*). Another way forward, therefore, consists in combining no-report paradigms (*Sergent et al., 2021*; *Tsuchiya et al., 2015*) with our EEG markers to examine whether local or global recurrent processing more accurately reflects consciousness in the absence of report.

## Methods

### Participants

Thirty-three participants took part in the first two sessions (independent EEG training set and practice). Three of them met the practice session's pre-established criteria for exclusion (see 'Procedure'). The remaining 30 participants (22±3-y-old, 10 men, two left-handed) took part in the final (main experimental) session. They all had normal or corrected-to-normal vision. The study was approved by the local ethics committee. Participants gave informed consent and received research credits or 15 euros per hour.

### Stimuli

The target stimulus set had a 2 (illusory Kanizsa triangle: present/absent)×2 (non-illusory triangle: present/absent)×2 (rotation: present/absent) design, resulting in eight stimuli (*Figure 1B*). Three aligned Pac-Man elements induced the Kanizsa illusion. The non-illusory triangle was present when the stimuli's three other elements (the 'two-legged white circles') were aligned. The controls for both the illusory and non-illusory triangles were created by rotating their elements by 90 degrees. Differences in local contrast were created by rotating the entire stimulus by 180 degrees. The targets spanned 7.5 degrees by 8.3 degrees of visual angle. The shortest distance between the edges of the three Pac-Man stimuli as well as between the edges of the three aligned two-legged white circles was 2.8 degrees of visual angle. Although neuronal responses to collinearity in primary visual cortex are most robust when this distance is smaller (*Kapadia et al., 1995*; *Kapadia et al., 2000*), longer-range lateral connections between neurons with similar orientation selectivity can span distances corresponding to visual angles considerably greater than 2.8 degrees (*Bosking et al., 1997*; *Stettler et al., 2002*).

The distractor stimulus set was the same as the target stimulus set, with two exceptions. First, the distractors were red instead of black. Second, the distractors' six elements were rotated by 180 degrees relative to the targets', so neither the illusory nor non-illusory triangle was ever present in the distractors. Masks consisted of six differently shaped elements, all capable of covering the targets' elements. Six masks were created by rotating the original mask five times by 60 degrees. They spanned 8.5 degrees by 9.1 degrees of visual angle. The fixation cross, which was always present, was adapted from *Thaler et al., 2013*.

### Procedure

The experiment consisted of three separate sessions conducted on different days: a 3 hr session to collect EEG data for the independent training set, a 1.5 hr practice session, and a 3 hr experimental session. Tasks were programmed in Presentation software (Neurobehavioral Systems) and displayed on a 23-inch, 60 Hz, 1920×1080 pixels monitor. On each trial of the experimental session, participants were shown two targets (T1 and T2) within a rapid serial visual presentation (RSVP) of distractors (*Figure 1C*). The targets and distractors had a stimulus onset asynchrony of 100 ms. T2 and distractors were presented for 17ms each. To improve the decoding analyses' training dataset, T1 was presented for 67 ms. The longer presentation duration facilitated attending to T1, which should result in greater deployment of attentional resources and thereby increase the size of the AB. T1 was preceded by five distractors, and T2 was followed by six distractors.

T2 visibility was manipulated in two ways, using a perceptual and an attentional manipulation (*Figure 1A*). The perceptual manipulation consisted of masking T2 with three masks, each presented for 17 ms with an interstimulus interval of 0 ms. The three masks were selected randomly, but all differed from each other. Half of the T2s were masked; for the other half, no masks were presented (unmasked condition). The attentional manipulation consisted of shortening the T1–T2 lag from a long interval of eight distractors (900 ms) to a short interval of one or two distractors (200 or 300 ms). Half the trials had a long lag, and the other half had a short lag. The short lag duration was determined for each participant individually during the training session. Short lags were expected to result in an AB. Participants were instructed to fixate on the fixation cross. After the RSVP, they indicated for each target whether it contained the Kanizsa illusion or not. For T2, participants simultaneously reported their confidence in their response: low, medium, or high, resulting in six response options. To get accurate ratings, participants first responded to T2 and then to T1. Response screens lasted until the response. In short, the experimental session had an

8 (T1 stimulus conditions)×8 (T2 stimulus conditions)×2 (masked/unmasked)×2 (short/long T1–T2 lag) task design, resulting in 256 conditions. Each condition was presented four times, totaling 1024 trials.

The experimental session was preceded by the practice session, in which participants were familiarized with the task. To proceed to the experimental session, participants had to score above 80% correct for both T1 and unmasked, long lag T2. One participant was excluded for failing to achieve this. The training session was also used to determine for each participant the duration of the short lag (200 or 300 ms T1–T2 interval) that induced the largest AB (lowest T2 accuracy) and that was used in the subsequent experimental session. Two participants were excluded due to their AB size falling below the predetermined criterion. Specifically, their T2 accuracy at both short lags did not exhibit a decrease of more than 5% compared to long lags.

One of the main goals of this study was to match perceptual performance between the perceptual and the attentional manipulation. We did this in two ways. First, during the training session, the matching was done by staircasing mask contrast using the weighted up-down method (*Kaernbach, 1991*). Contrast levels ranged from 0 (black) to 255 (white). Mask contrast started at level 220. Each correct response made the task more difficult: masks got darker by downward step size $S_{down}$. Each incorrect response made the task easier: masks got lighter by upward step size $S_{up}$. Step sizes were determined by $S_{up} \times p = S_{down} \times (1 - p)$, where $p$ is the accuracy at short lags. The smallest step size was always nine contrast levels. A reversal is making a mistake after a correct response, or vice versa. The staircase ended after 25 reversals. The mask contrast with which the experimental session started was the average contrast level of the last 20 reversals. Second, during the experimental session, after every 32 masked trials, mask contrast could be updated in accordance with our goal to match accuracy over participants, while also matching accuracy within participants as well as possible.

To ensure that confidence ratings for these matched conditions (masked, long lag and unmasked, short lag) were not influenced by participants anchoring their confidence ratings to the very easy and very difficult unmatched conditions (no and both manipulations, respectively), one type of block only contained the matched conditions, while the other block type contained the two remaining, unmatched conditions (masked, short lag and unmasked, long lag). To ensure every confidence rating would have enough trials for creating receiver operating characteristic curves, participants were instructed to distribute their responses evenly over all ratings within a block. Participants received feedback about the distribution of their responses. The mask contrasts from a performance matched block were used in the subsequent non-performance matched block to ensure that masking remained orthogonal to the AB manipulation. The experimental session, therefore, always started with a performance matched block.

We wanted to compare illusory triangle decoding to non-illusory triangle decoding. However, during the experimental session, the non-illusory triangle was never task-relevant, only the illusory one was. During the independent EEG classification training session, we, therefore, made each visual feature, one after the other, task-relevant. A target was presented for 33 ms every 900–1100 ms (*Figure 3A*). Participants had to fixate on the fixation cross and indicate whether the current task-relevant feature was absent or present. For each feature, each target was presented 64 times, totaling 512 trials. The order of the task-relevant features was counterbalanced over participants. For all sessions, response button mapping was counterbalanced within tasks.

## Behavioral analysis

To quantify perceptual performance, we constructed receiver operating characteristic (ROC) curves by plotting objective hit rates against objective false alarm rates. We used the six response options to get five inflection points (*Green and Swets, 1966*). We also quantified metacognitive sensitivity: the ability to know whether you were right or wrong. Performance is high when you are confident in objectively correct responses and not confident in objectively incorrect responses. We again constructed ROC curves, now by plotting the rate of high-confidence correct responses (subjective hit rates) against the rate of high-confidence incorrect responses (subjective false alarm rates). We used the three confidence ratings to get two inflection points. To ensure that T1 was attended, trials with incorrect T1 responses were excluded. Frequentist and Bayesian repeated measures ANOVAs and t-tests were used to test the differences between experimental conditions. Bayesian statistics were calculated in JASP (*JASP Team, 2024*) with default prior scales (Cauchy distribution, scale 0.707).

## EEG recording and preprocessing

EEG was recorded at 1024 Hz using a 64-channel ActiveTwo system (BioSemi). Four electrooculographic (EOG) electrodes measured horizontal and vertical eye movements. The data were analyzed with MATLAB (MathWorks). For most of the preprocessing steps, EEGLAB was used (*Delorme and Makeig, 2004*). The data were re-referenced to the earlobes. Poor channels were interpolated. High-pass filtering can cause artifacts in decoding analyses; we, therefore, removed slow drifts using trial-masked robust detrending (*van Driel et al., 2021*). Each target was epoched from –250–1000 ms relative to target onset. To improve the results from the independent component analysis (ICA), baseline correction was applied using the whole epoch as baseline (*Groppe et al., 2009*). ICA was used to remove blinks. Blink components were removed manually. Baseline correction was applied, now using a –250–0 ms window relative to target onset. Trials with values outside a –300–300 microvolts range were removed. We used an adapted version of FieldTrip's ft_artifact_zvalue function to detect and remove trials with muscle artifacts (*Oostenveld et al., 2011*). As in the behavioral analyses, trials with incorrect T1 responses were excluded. Finally, the data were downsampled to 128 Hz.

## Multivariate pattern analyses

We decoded the different visual features (local contrast, non-illusory triangle, and illusory Kanizsa triangle, respectively; *Figure 1B*) using the Amsterdam Decoding and Modeling (ADAM) toolbox (*Fahrenfort et al., 2018*). For each participant and each visual feature, a linear discriminant classifier was trained on the T1 data and tested on each condition of the T2 data. The classifier was trained to discriminate between the feature's (e.g. the illusory triangle's) absence and presence based on the preprocessed EEG activity across all electrodes. AUC was again used as the performance measure. This procedure was executed for every time sample in a trial, yielding classification performance over time. For the time samples from –100–700 ms relative to target onset, we used a two-sided t-test to evaluate whether classifier performance differed from chance. We used cluster-based permutation testing (1000 iterations at a threshold of 0.05) to correct for multiple comparisons (*Maris and Oostenveld, 2007*). To obtain topographic maps showing the neural sources of the classifier performance, we multiplied the classifier weights with the data covariance matrix, yielding covariance/class separability maps (*Haufe et al., 2014*).

In the decoding analyses described in the results, we applied 'diagonal decoding:' classifiers were tested on the same time sample they were trained on. We did the same analyses again, now by applying 'off-diagonal decoding:' classifiers trained on a particular time point are tested on all time points (*King and Dehaene, 2014*). Off-diagonal decoding allowed us to explore whether patterns of activity during the time windows of interest were stable over time (*Figure 2—figure supplement 2*), which is especially interesting for illusion decoding. For the illusory triangle, classifiers were trained on the 200–250 ms window and then averaged, and then decoding AUC was plotted from –100–700 ms off-diagonally. Note that for this analysis, late off-diagonal decoding (e.g. in the 375–475 ms time-window shown in panel C) is thought to reflect (longer-lasting) local recurrent interactions as well (*Alilović et al., 2023*). A similar off-diagonal analysis was done for the local contrast trained on the 75–95 ms time-window.

To distinguish between collinearity-only and illusion-specific processing, we trained classifiers on independent data based on collinearity-only (the non-illusory triangle was task-relevant) or collinearity-plus-illusion (the illusory triangle was task-relevant) and then decoded the Kanizsa illusion in T2s of the main RSVP task. The rationale for this analysis is that collinearity is present both when the Pac-Man stimuli align to form the illusory Kanizsa triangle and when the two-legged white circles align to form a non-illusory triangle, but only in the case of the Kanizsa triangle do participants experience an illusion. The comparison of T2 illusion decoding between the classifiers trained on the illusion and on collinearity-only in the training set may, therefore, isolate illusion-specific (likely involving feedback processing) from basic collinearity processing (likely involving lateral connections; *Figure 3*). In **Appendix 1** and *Figure 3—figure supplement 2*, we compare the independent training set to the training set used for the main analyses, the T1 data from the RSVP task.

As described in **Appendix 1** as well, a 10-fold cross-validation scheme was applied to the data from the independent training set to decode local contrast. Individual participants' data were split into ten equal-sized folds after randomizing the task's trial order. A classifier was then trained on ninefolds and tested on the tenth one, ensuring independence of the training and testing sets. This procedure

was repeated until each fold served as the test set once. Classifier performance, AUC, was averaged across all ten iterations (*Figure 3—figure supplement 1*).

As in the behavioral analyses, frequentist and Bayesian repeated measures ANOVAs and t-tests were used to test the differences between experimental conditions.

## Additional information

### Competing interests
Simon van Gaal: Reviewing editor, eLife. The other authors declare that no competing interests exist.

### Funding

| Funder | Grant reference number | Author |
| --- | --- | --- |
| HORIZON EUROPE European Research Council | 10.3030/715605 | Simon van Gaal |

The funders had no role in study design, data collection and interpretation, or the decision to submit the work for publication.

### Author contributions
Samuel Noorman, Conceptualization, Data curation, Formal analysis, Investigation, Visualization, Methodology, Writing – original draft, Project administration, Writing – review and editing; Timo Stein, Supervision, Writing – original draft, Writing – review and editing; Johannes Jacobus Fahrenfort, Conceptualization, Supervision, Methodology, Writing – review and editing; Simon van Gaal, Conceptualization, Supervision, Funding acquisition, Writing – original draft, Project administration, Writing – review and editing

### Author ORCIDs
Samuel Noorman ⓘ https://orcid.org/0000-0003-0953-1070
Timo Stein ⓘ https://orcid.org/0000-0002-8484-0933
Johannes Jacobus Fahrenfort ⓘ https://orcid.org/0000-0002-9025-3436
Simon van Gaal ⓘ https://orcid.org/0000-0001-6628-4534

### Ethics
Human subjects: Informed consent and consent to publish was obtained from all participants. All procedures were approved by the ethics board of the Psychology department of the University of Amsterdam, under project number 2020-BC-12346.

Reviewer #1 (Public review): https://doi.org/10.7554/eLife.97900.3.sa1
Reviewer #2 (Public review): https://doi.org/10.7554/eLife.97900.3.sa2
Reviewer #3 (Public review): https://doi.org/10.7554/eLife.97900.3.sa3
Author response https://doi.org/10.7554/eLife.97900.3.sa4

## Additional files

### Supplementary files
MDAR checklist

### Data availability
Data related to this paper is available at UvA/AUAS figshare. Code related to this paper is available at UvA/AUAS figshare.

The following datasets were generated:

| Author(s) | Year | Dataset title | Dataset URL | Database and Identifier |
|---|---|---|---|---|
| Noorman SG | 2024 | Data: Perceptual and attentional impairments of conscious access involve distinct neural mechanisms despite equal task performance | https://doi.org/10.21942/uva.28049231.v2 | UvA/AUAS figshare, 10.21942/uva.28049231.v2 |
| Noorman SG | 2024 | Code: Perceptual and attentional impairments of conscious access involve distinct neural mechanisms despite equal task performance | https://doi.org/10.21942/uva.28049423.v1 | UvA/AUAS figshare, 10.21942/uva.28049423.v1 |

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

## Appendix 1

The experimental blocks of the independent training set differed from the main rapid serial visual presentation (RSVP) task in task context, trial design, and stimulus timing (stimuli were presented briefly and in isolation) (*Figure 1C* vs. *Figure 3A*) and were collected on a different day, with fewer trials. This led to overall slightly lower decoding performance compared to the main analyses based on training classifiers on the T1 data (*Figure 2—figure supplement 1*, top vs. bottom panels), likely reflecting these design differences and greater differences between training and test data with regard to conscious access and working memory demands. Despite these differences, we successfully used these independent classifiers to replicate (*Figure 3—figure supplement 2*) and expand upon (*Figure 3*) the results from our main analyses.

*Figure 3* shows the comparison of T2 illusory triangle decoding between the classifiers trained on the illusory (green lines) and non-illusory triangle in the independent training set (purple lines), because this comparison may isolate illusion-specific processing from basic collinearity-only processing. For this analysis approach to be valid, collinearity-only processing should be similar for the illusory and the non-illusory triangle. To demonstrate that this was indeed the case, we reversed the original cross-feature-decoding scheme: Instead of testing on the illusory triangle, we tested on the non-illusory triangle in T2s, while using the same classifiers that had been trained on either the non-illusory or illusory triangle using the independent training data (*Figure 3—figure supplement 3A*, green and purple lines). If collinearity-only processing did not differ between the two types of triangles, there should now be no difference in decoding performance between the classifiers, which is what we found for both the 140–190 ($t_{29}=-0.90$, p=0.378, $BF_{01}=3.56$) and the 200–250ms window ($t_{29}=0.69$, p=0.493, $BF_{01}=4.12$; *Figure 3—figure supplement 3B*).

We also sought to demonstrate that training on the non-illusory triangle and testing on the illusory triangle in T2s, as reported in the main manuscript, indeed reflected collinearity processing (i.e. the visual properties shared by the two types of triangles, their shape), and no other irrelevant, e.g., task- or attention-related processes. For this purpose, we also trained a classifier on local contrast using the independent training data and tested on the illusory triangle. Local contrast is fully orthogonal to the illusion and has nothing (visual) in common with the illusion, so that any successful decoding would have to reflect non-visual, irrelevant processes. However, we found that this decoding scheme's classifier performance never reached significance (p>0.05).

Finally, for local contrast decoding, we examined whether the fact that local contrast was not task-relevant or explicitly attended during the main RSVP task could have influenced our findings. We addressed this question by leveraging the manipulation of task-relevance in the independent training set, where either local contrast ('Was the target rotated 180 degrees?'), the non-illusory triangle ('Did the target contain a non-illusory triangle?'), or the illusory triangle ('Did the target contain an illusory triangle?') was task-relevant. For each task-relevance condition, local contrast was decoded using a tenfold cross-validation scheme, and the resulting 75–95 ms time window was again averaged (*Figure 3—figure supplement 1A*). Surprisingly, we found that classifier performance was somewhat better when local contrast was task-irrelevant ($F_{1,29}=5.30$, p=0.008, $BF_{10}=5.39$). However, when we repeated this analysis using data from an independent study with virtually the same task design, we found that task-relevance had no significant effect on local contrast decoding ($F_{1,29}=1.42$, p=0.251, $BF_{01}=3.17$; *Figure 3—figure supplement 1B*). Local contrast decoding thus appeared largely unaffected by task-relevance.

Using the independent training set, we also replicated the observation that the two consciousness manipulations left local contrast decoding largely intact. For this control analysis, we trained the classifier on the independent training set in which local contrast was task-relevant (*Figure 3B*, light blue lines). As for our previous main analyses, we averaged the 75–95 ms time window, as this window again contained the decoding peak with occipital topography (*Figure 2—figure supplement 1A*, bottom). Similar to our main analyses, another rm ANOVA with the factors masking (present/absent) and T1-T2 lag (short/long) showed that neither masking ($F_{1,29}=0.97$, p=0.334, $BF_{01}=2.90$) nor the T1-T2 lag ($F_{1,29}=0.72$, p=0.403, $BF_{01}=3.54$) had a significant effect on local contrast decoding, and a paired t-test revealed no evidence of a significant difference between the two performance matched conditions (masked vs. AB condition, $t_{29}=1.20$, p=0.240, $BF_{01}=2.68$; *Figure 3C*, '75–95 ms').

