## [Editor Report · eLife Assessment]

This **important** study provides new insights into the mechanisms that underlie perceptual and attentional impairments of conscious access. The paper presents **convincing** evidence of a dissociation between the early stages of low-level perception, which are impermeable to perceptual or attentional impairments, and subsequent stages of visual integration which are susceptible to perceptual impairment but resilient to attentional manipulations. This study will be of interest to scientists working on visual perception and consciousness.

---

## [Referee Report · Reviewer #1 (Public review)]

Summary:

In this work, Noorman and colleagues test the predictions of the "four-stage model" of consciousness by combining psychophysics and scalp EEG in humans. The study relies on an elegant experimental design to investigate the respective impact of attentional and perceptual blindness on visual processing.

The study is very well summarised, the text is clear and the methods seem sound. Overall, a very solid piece of work. I haven't identified any major weaknesses. Below I raise a few questions of interpretation that may possibly be the subject of a revision of the text.

(1) The perceptual performance on Fig1D appears to show huge variation across participants, with some participants at chance levels and others with performance > 90% in the attentional blink and/or masked conditions. This seems to reveal that the procedure to match performance across participants was not very successful. Could this impact the results? The authors highlight the fact that they did not resort to post-selection or exclusion of participants, but at the same time do not discuss this equally important point.

(2) In the analysis on collinearity and illusion-specific processing, the authors conclude that the absence of a significant effect of training set demonstrates collinearity-only processing. I don't think that this conclusion is warranted: as the illusory and non-illusory share the same shape, so more elaborate object processing could also be occuring. Please discuss.

(3) Discussion, lines 426-429: It is stated that the results align with the notion that processes of perceptual segmentation and organization represent the mechanism of conscious experience. My interpretation of the results is that they show the contrary: for the same visibility level in the attentional blind or masking conditions, these processes can be implicated or not, which suggests a role during unconscious processing instead.

(4) The two paradigms developed here could be used jointly to highlight non-idiosyncratic NCCs, i.e. EEG markers of visibility or confidence that generalise regardless of the method used. Have the authors attempted to train the classifier on one method and apply it to another (e.g. AB to masking and vice versa)? What perceptual level is assumed to transfer?

(4) How can the results be integrated with the attentional literature showing that attentional filters can be applied early in the processing hierarchy?

Comments on revisions:

I'm very pleased with the responses to my previous comments, and congratulate the authors on this excellent piece of work.

---

## [Referee Report · Reviewer #2 (Public review)]

Summary:

This is a very elegant and important EEG study that unifies within a single set of behaviorally equated experimental conditions conscious access (and therefore also conscious access failures) during visual masking and attentional blink (AB) paradigms in humans. By a systematic and clever use of multivariate pattern classifiers across conditions, they could dissect, confirm, and extend a key distinction (initially framed within the GNWT framework) between 'subliminal' and 'pre-conscious' unconscious levels of processing. In particular, the authors could provide strong evidence to distinguish here within the same paradigm these two levels of unconscious processing that precede conscious access : (i) an early (< 80ms) bottom-up and local (in brain) stage of perceptual processing ('local contrast processing') that was preserved in both unconscious conditions, (ii) a later stage and more integrated processing (200-250ms) that was impaired by masking but preserved during AB. On the basis of preexisting studies and theoretical arguments, they suggest that this later stage could correspond to lateral and local recurrent feedback processes. Then, the late conscious access stage appeared as a P3b-like event.

Strengths:

The methodology and analyses are strong and valid. This work adds an important piece in the current scientific debate about levels of unconscious processing and specificities of conscious access in relation to feed-forward, lateral, and late brain-scale top-down recurrent processing.

Comments on revisions:

I congratulate the authors for the quality of their revised ms. They convincingly addressed each of the issues raised in my previous review.

---

## [Referee Report · Reviewer #3 (Public review)]

Summary:

This work aims to investigate how perceptual and attentional processes affect conscious access in humans. By using multivariate decoding analysis of electroencephalography (EEG) data, the authors explored the neural temporal dynamics of visual processing across different levels of complexity (local contrast, collinearity, and illusory perception). This is achieved by comparing the decidability of an illusory percept in matched conditions of perceptual (i.e., degrading the strength of sensory input using visual masking) and attentional impairment (i.e., impairing top-down attention using attentional blink, AB). The decoding results reveal three distinct temporal responses associated with the three levels of visual processing. Interestingly, the early stage of local contrast processing remains unaffected by both masking and AB. However, the later stage of collinearity and illusory percept processing are impaired by the perceptual manipulation but remained unaffected by the attentional manipulation. These findings contribute to the understanding of the unique neural dynamics of perceptual and attentional functions and how they interact with the different stages of conscious access.

Strengths:

The study investigates perceptual and attentional impairments across multiple levels of visual processing in a single experiment. Local contrast, collinearity, and illusory perception were manipulated using different configurations of the same visual stimuli. This clever design allows for the investigation of different levels of visual processing under similar low-level conditions.

Moreover, behavioural performance was matched between perceptual and attentional manipulations. One of the main problems when comparing perceptual and attentional manipulations on conscious access is that they tend to impact performance at different levels, with perceptual manipulations like masking producing larger effects. The study utilizes a staircasing procedure to find the optimal contrast of the mask stimuli to produce a performance impairment to the illusory perception comparable to the attentional condition, both in terms of perceptual performance (i.e., indicating whether the target contained the Kanizsa illusion) and metacognition (i.e., confidence in the response).

The results show a clear dissociation between the three levels of visual processing in terms of temporal dynamics. Local contrast was represented at an early stage (~80 ms), while collinearity and illusory perception were associated with later stages (~200-250 ms). Furthermore, the results provide clear evidence in support of a dissociation between the effects of perceptual and attentional processes on conscious access: while the former affected both neuronal correlates of collinearity and illusory perception, the latter did not have any effect on the processing of the more complex visual features involved in the illusion perception.

Weaknesses:

The design of the study and the results presented are very similar to those in Fahrenfort et al. (2017), reducing its novelty. Similar to the current study, Fahrenfort et al. (2017) tested the idea that if both masking and AB impact perceptual integration, they should affect the neural markers of perceptual integration in a similar way. They found that behavioural performance (hit/false alarm rate) was affected by both masking and AB, even though only the latter was significant in the unmasked condition. In contrast, an early classification peak was exclusively affected by masking. A later classification peak mirrored the behavioural findings, with classification performance impacted by both masking and AB.

The interpretation of the results primarily relies on the recurrent processing theory of consciousness (Lamme, 2020), which lead to the assumption that local contrast and illusory perception reflect feedforward and (lateral and feedback) recurrent connections, respectively. It should be mentioned, however, that this theoretical prediction is not directly tested in the study. Moreover, the evidence for the dissociation between illusion and collinearity in terms of lateral and feedback connections seems at least limited. For instance, Kok et al. (2016) found that, whereas bottom-up stimulation activated all cortical layers, feedback activity induced by illusory figures led to a selective activation of the deep layers. Lee & Nguyen (2001), instead, found that V1 neurons respond to illusory contours of the Kanizsa figures, particularly in the superficial layers. Although both studies reference feedback connections, neither provides clear evidence for the involvement of lateral connections.

The evidence in favour of primarily lateral connections driving collinearity seems mixed as well. On one hand, Liang et al. (2017) showed that feedback and lateral connections closely interact to mediate image grouping and segmentation. On the other hand, Stettler et al. (2002) showed that, whereas the intrinsic connections link similarly oriented domains in V1, V2 to V1 feedback displays no such specificity. Additionally, the other studies cited in the manuscript focused solely on lateral connections without examining feedback pathways, making it challenging to draw definitive conclusions.

Comments on revisions:

The authors have thoroughly addressed all my comments and provided comprehensive responses to each point raised.

---

## [Author Response]

The following is the authors’ response to the original reviews.

**Reviewer #1 (Public Review):**
Summary:In this work, Noorman and colleagues test the predictions of the "four-stage model" of consciousness by combining psychophysics and scalp EEG in humans. The study relies on an elegant experimental design to investigate the respective impact of attentional and perceptual blindness on visual processing.The study is very well summarised, the text is clear and the methods seem sound. Overall, a very solid piece of work. I haven't identified any major weaknesses. Below I raise a few questions of interpretation that may possibly be the subject of a revision of the text.

We thank the reviewer for their positive assessment of our work and for their extremely helpful and constructive comments that helped to significantly improve the quality of our manuscript.

(1) The perceptual performance on Fig1D appears to show huge variation across participants, with some participants at chance levels and others with performance > 90% in the attentional blink and/or masked conditions. This seems to reveal that the procedure to match performance across participants was not very successful. Could this impact the results? The authors highlight the fact that they did not resort to postselection or exclusion of participants, but at the same time do not discuss this equally important point.

Performance was indeed highly variable between observers, as is commonly found in attentional-blink (AB) and masking studies. For some observers, the AB pushes performance almost to chance level, whereas for others it has almost no effect. A similar effect can be seen in masking. We did our best to match accuracy over participants, while also matching accuracy within participants as well as possible, adjusting mask contrast manually during the experimental session. Naturally, those that are strongly affected by masking need not be the same participants as those that are strongly affected by the AB, given the fact that they rely on different mechanisms (which is also one of the main points of the manuscript). To answer the research question, what mattered most was that at the group-level, performance was well matched between the two key conditions. As all our statistical inferences, both for behavior and EEG decoding, rest on this group level. We do not think that variability at the individualsubject level detracts from this general approach.

In the Results, we added that our goal was to match performance across participants:

“Importantly, mask contrast in the masked condition was adjusted using a staircasing procedure to match performance in the AB condition, ensuring comparable perceptual performance in the masked and the AB condition across participants (see Methods for more details).”

In the Methods, we added:

“Second, during the experimental session, after every 32 masked trials, mask contrast could be manually updated in accordance with our goal to match accuracy over participants, while also matching accuracy within participants as well as possible.”

(2) In the analysis on collinearity and illusion-specific processing, the authors conclude that the absence of a significant effect of training set demonstrates collinearity-only processing. I don't think that this conclusion is warranted: as the illusory and nonillusory share the same shape, so more elaborate object processing could also be occurring. Please discuss.

We agree with this qualification of our interpretation, and included the reviewer’s account as an alternative explanation in the Discussion section:

“It should be noted that not all neurophysiological evidence unequivocally links processing of collinearity and of the Kanizsa illusion to lateral and feedback processing, respectively (Angelucci et al., 2002; Bair et al., 2003; Chen et al., 2014), so that overlap in decoding the illusory and non-illusory triangle may reflect other mechanisms, for example feedback processes representing the triangular shapes as well.”

(3) Discussion, lines 426-429: It is stated that the results align with the notion that processes of perceptual segmentation and organization represent the mechanism of conscious experience. My interpretation of the results is that they show the contrary: for the same visibility level in the attentional blind or masking conditions, these processes can be implicated or not, which suggests a role during unconscious processing instead.

We agree with the reviewer that the interpretation of this result depends on the definition of consciousness that one adheres to. If one takes report as the leading metric for consciousness (=conscious access), one can indeed conclude that perceptual segmentation/organization can also occur unconsciously. However, if the processing that results in the qualitative nature of an image (rather than whether it is reported) is taken as leading – such as the processing that results in the formation of an illusory percept – (=phenomenal) the conclusion can be quite different. This speaks to the still ongoing debate regarding the existence of phenomenal vs access consciousness, and the literature on no-report paradigms amongst others (see last paragraph of the discussion). Because the current data do not speak directly to this debate, we decided to remove the sentence about “conscious experience”, and edited this part of the manuscript (also addressing a comment about preserved unconscious processing during masking by Reviewer 2) by limiting the interpretation of unconscious processing to those aspects that are uncontroversial:

“Such deep feedforward processing can be sufficient for unconscious high-level processing, as indicated by a rich literature demonstrating high-level (e.g., semantic) processing during masking (Kouider & Dehaene, 2007; Van den Bussche et al., 2009; van Gaal & Lamme, 2012). Thus, rather than enabling deep unconscious processing, preserved local recurrency during inattention may afford other processing advantages linked to its proposed role in perceptual integration (Lamme, 2020), such as integration of stimulus elements over space or time.”

(4) The two paradigms developed here could be used jointly to highlight nonidiosyncratic NCCs, i.e. EEG markers of visibility or confidence that generalise regardless of the method used. Have the authors attempted to train the classifier on one method and apply it to another (e.g. AB to masking and vice versa)? What perceptual level is assumed to transfer?

To avoid issues with post-hoc selection of (visible vs. invisible) trials (discussed in the Introduction), we did not divide our trials into conscious and unconscious trials, and thus did not attempt to reveal NCCs, or NCCs generalizing across the two paradigms. Note also that this approach alone would not resolve the debate regarding the ‘true’ NCC as it hinges on the operational definition of consciousness one adheres to; also see our response to the previous point the reviewer raised. Our main analysis revealed that the illusory triangle could be decoded with above-chance accuracy during both masking and the AB over extended periods of time with similar topographies (Fig. 2B), so that significant cross-decoding would be expected over roughly the same extended period of time (except for the heightened 200-250 ms peak). However, as our focus was on differences between the two manipulations and because we did not use post-hoc sorting of trials, we did not add these analyses.

(5) How can the results be integrated with the attentional literature showing that attentional filters can be applied early in the processing hierarchy?

Compared to certain manipulations of spatial attention, the AB phenomenon is generally considered to represent an instance of “late” attentional filtering. In the Discussion section we included a paragraph on classic load theory, where early and late filtering depend on perceptual and attentional load. Just preceding this paragraph, we added this:

“Clearly, these findings do not imply that unconscious high-level (e.g., semantic) processing can only occur during inattention, nor do they necessarily generalize to other forms of inattention. Indeed, while the AB represents a prime example of late attentional filtering, other ways of inducing inattention or distraction (e.g., by manipulating spatial attention) may filter information earlier in the processing hierarchy (e.g., Luck & Hillyard, 1994 vs. Vogel et al., 1998).”

**Reviewer #2 (Public Review):**
Summary:This is a very elegant and important EEG study that unifies within a single set of behaviorally equated experimental conditions conscious access (and therefore also conscious access failures) during visual masking and attentional blink (AB) paradigms in humans. By a systematic and clever use of multivariate pattern classifiers across conditions, they could dissect, confirm, and extend a key distinction (initially framed within the GNWT framework) between 'subliminal' and 'pre-conscious' unconscious levels of processing. In particular, the authors could provide strong evidence to distinguish here within the same paradigm these two levels of unconscious processing that precede conscious access : (i) an early (< 80ms) bottom-up and local (in brain) stage of perceptual processing ('local contrast processing') that was preserved in both unconscious conditions, (ii) a later stage and more integrated processing (200-250ms) that was impaired by masking but preserved during AB. On the basis of preexisting studies and theoretical arguments, they suggest that this later stage could correspond to lateral and local recurrent feedback processes. Then, the late conscious access stage appeared as a P3b-like event.Strengths:The methodology and analyses are strong and valid. This work adds an important piece in the current scientific debate about levels of unconscious processing and specificities of conscious access in relation to feed-forward, lateral, and late brain-scale top-down recurrent processing.Weaknesses:- The authors could improve clarity of the rich set of decoding analyses across conditions.- They could also enrich their Introduction and Discussion sections by taking into account the importance of conscious influences on some unconscious cognitive processes (revision of traditional concept of 'automaticity'), that may introduce some complexity in Results interpretation- They should discuss the rich literature reporting high-level unconscious processing in masking paradigms (culminating in semantic processing of digits, words or even small group of words, and pictures) in the light of their proposal (deeper unconscious processing during AB than during masking).

We thank the reviewer for their positive assessment of our study and for their insightful comments and helpful suggestions that helped to significantly strengthen our paper. We provide a more detailed point-by-point response in the “recommendations for the authors” section below. In brief, we followed the reviewer’s suggestions and revised the Results/Discussion to include references to influences on unconscious processes and expanded our discussion of unconscious effects during masking vs. AB.

**Reviewer #3 (Public Review):**
Summary:This work aims to investigate how perceptual and attentional processes affect conscious access in humans. By using multivariate decoding analysis of electroencephalography (EEG) data, the authors explored the neural temporal dynamics of visual processing across different levels of complexity (local contrast, collinearity, and illusory perception). This is achieved by comparing the decidability of an illusory percept in matched conditions of perceptual (i.e., degrading the strength of sensory input using visual masking) and attentional impairment (i.e., impairing topdown attention using attentional blink, AB). The decoding results reveal three distinct temporal responses associated with the three levels of visual processing. Interestingly, the early stage of local contrast processing remains unaffected by both masking and AB. However, the later stage of collinearity and illusory percept processing are impaired by the perceptual manipulation but remain unaffected by the attentional manipulation. These findings contribute to the understanding of the unique neural dynamics of perceptual and attentional functions and how they interact with the different stages of conscious access.Strengths:The study investigates perceptual and attentional impairments across multiple levels of visual processing in a single experiment. Local contrast, collinearity, and illusory perception were manipulated using different configurations of the same visual stimuli. This clever design allows for the investigation of different levels of visual processing under similar low-level conditions.Moreover, behavioural performance was matched between perceptual and attentional manipulations. One of the main problems when comparing perceptual and attentional manipulations on conscious access is that they tend to impact performance at different levels, with perceptual manipulations like masking producing larger effects. The study utilizes a staircasing procedure to find the optimal contrast of the mask stimuli to produce a performance impairment to the illusory perception comparable to the attentional condition, both in terms of perceptual performance (i.e., indicating whether the target contained the Kanizsa illusion) and metacognition (i.e., confidence in the response).The results show a clear dissociation between the three levels of visual processing in terms of temporal dynamics. Local contrast was represented at an early stage (~80 ms), while collinearity and illusory perception were associated with later stages (~200-250 ms). Furthermore, the results provide clear evidence in support of a dissociation between the effects of perceptual and attentional processes on conscious access: while the former affected both neuronal correlates of collinearity and illusory perception, the latter did not have any effect on the processing of the more complex visual features involved in the illusion perception.Weaknesses:The design of the study and the results presented are very similar to those in Fahrenfort et al. (2017), reducing its novelty. Similar to the current study, Fahrenfort et al. (2017) tested the idea that if both masking and AB impact perceptual integration, they should affect the neural markers of perceptual integration in a similar way. They found that behavioural performance (hit/false alarm rate) was affected by both masking and AB, even though only the latter was significant in the unmasked condition. An early classification peak was instead only affected by masking. However, a late classification peak showed a pattern similar to the behavioural results, with classification affected by both masking and AB.The interpretation of the results mainly centres on the theoretical framework of the recurrent processing theory of consciousness (Lamme, 2020), which lead to the assumption that local contrast, collinearity, and the illusory perception reflect feedforward, local recurrent, and global recurrent connections, respectively. It should be mentioned, however, that this theoretical prediction is not directly tested in the study. Moreover, the evidence for the dissociation between illusion and collinearity in terms of lateral and feedback connections seems at least limited. For instance, Kok et al. (2016) found that, whereas bottom-up stimulation activated all cortical layers, feedback activity induced by illusory figures led to a selective activation of the deep layers. Lee & Nguyen (2001), instead, found that V1 neurons respond to illusory contours of the Kanizsa figures, particularly in the superficial layers. They all mention feedback connections, but none seem to point to lateral connections.Moreover, the evidence in favour of primarily lateral connections driving collinearity seems mixed as well. On one hand, Liang et al. (2017) showed that feedback and lateral connections closely interact to mediate image grouping and segmentation. On the other hand, Stettler et al. (2002) showed that, whereas the intrinsic connections link similarly oriented domains in V1, V2 to V1 feedback displays no such specificity. Furthermore, the other studies mentioned in the manuscript did not investigate feedback connections but only lateral ones, making it difficult to draw any clear conclusions.

We thank the reviewer for their careful review and positive assessment of our study, as well as for their constructive criticism and helpful suggestions. We provide a more detailed point-by-point response in the “recommendations for the authors” section below. In brief, we addressed the reviewer’s comments and suggestions by better relating our study to Fahrenfort et al.’s (2017) paper and by highlighting the limitations inherent in linking our findings to distinct neural mechanisms (in particular, to lateral vs. feedback connections).

**Recommendations for the authors:**

**Reviewer #1 (Recommendations For The Authors):**
- Methods: it states that "The distance between the three Pac-Man stimuli as well as between the three aligned two-legged white circles was 2.8 degrees of visual angle". It is unclear what this distance refers to. Is it the shortest distance between the edges of the objects?

It is indeed the shortest distance between the edges of the objects. This is now included in the Methods.

- Methods: It's unclear to me if the mask updating procedure during the experimental session was based on detection rate or on the perceptual performance index reported on Fig1D. Please clarify.

It was based on accuracy calculated over 32 trials. We have included this information in the Methods.

- Methods and Results: I did not understand why the described procedure used to ensure that confidence ratings are not contaminated by differences in perceptual performance was necessary. To me, it just seems to make the "no manipulations" and "both manipulations" less comparable to the other 2 conditions.

To calculate accurate estimates of metacognitive sensitivity for the two matched conditions, we wanted participants to make use of the full confidence scale (asking them to distribute their responses evenly over all ratings within a block). By mixing all conditions in the same block, we would have run the risk of participants anchoring their confidence ratings to the unmatched very easy and very difficult conditions (no and both manipulations condition). We made this point explicit in the Results section and in the Methods section:

“To ensure that the distribution of confidence ratings in the performancematched masked and AB condition was not influenced by participants anchoring their confidence ratings to the unmatched very easy and very difficult conditions (no and both manipulations condition, respectively), the masked and AB condition were presented in the same experimental block, while the other block type included the no and both manipulations condition.”

“To ensure that confidence ratings for these matched conditions (masked, long lag and unmasked, short lag) were not influenced by participants anchoring their confidence ratings to the very easy and very difficult unmatched conditions (no and both manipulations, respectively), one type of block only contained the matched conditions, while the other block type contained the two remaining, unmatched conditions (masked, short lag and unmasked, long lag).”

- Methods: what priors were used for Bayesian analyses?

Bayesian statistics were calculated in JASP (JASP Team, 2024) with default prior scales (Cauchy distribution, scale 0.707). This is now added to the Methods.

- Results, line 162: It states that classifiers were applied on "raw EEG activity" but the Methods specify preprocessing steps. "Preprocessed EEG activity" seems more appropriate.

We changed the term to “preprocessed EEG activity” in the Methods and to “(minimally) preprocessed EEG activity (see Methods)” in the Results, respectively.

- Results, line 173: The effect of masking on local contrast decoding is reported as "marginal". If the alpha is set at 0.05, it seems that this effect is significant and should not be reported as marginal.

We changed the wording from “marginal” to “small but significant.”

- Fig1: The fixation cross is not displayed.

Because adding the fixation cross would have made the figure of the trial design look crowded and less clear, we decided to exclude it from this schematic trial representation. We are now stating this also in the legend of figure 1.

- Fig 3A: In the upper left panel, isn't there a missing significant effect of the "local contrast training and testing" condition in the first window? If not, this condition seems oddly underpowered compared to the other two conditions.

Thanks for the catch! The highlighting in bold and the significance bar were indeed lacking for this condition in the upper left panel (blue line). We corrected the figure in our revision.

- Supplementary text and Fig S6: It is unclear to me why the two control analyses (the black lines vs. the green and purple lines) are pooled together in the same figure. They seem to test for different, non-comparable contrasts (they share neither training nor testing sets), and I find it confusing to find them on the same figure.

We agree that this may be confusing, and deleted the results from one control analysis from the figure (black line, i.e., training on contrast, testing on illusion), as the reviewer correctly pointed out that it displayed a non-comparable analysis. Given that this control analysis did not reveal any significant decoding, we now report its results only in the Supplementary text.

- Fig S6: I think the title of the legend should say testing on the non-illusory triangle instead of testing on the illusory triangle to match the supplementary text.

This was a typo – thank you! Corrected.

**Reviewer #2 (Recommendations For The Authors):**
Issue #1: One key asymmetry between the three levels of T2 attributes (i.e.: local contrast; non-illusory triangle; illusory Kanisza triangle) is related to the top-down conscious posture driven by the task that was exclusively focusing on the last attribute (illusory Kanisza triangle). Therefore, any difference in EEG decoding performance across these three levels could also depend to this asymmetry. For instance, if participants were engaged to report local contrast or non-illusory triangle, one could wonder if decoding performance could differ from the one used here. This potential confound was addressed by the authors by using decoders trained in different datasets in which the main task was to report one the two other attributes. They could then test how classifiers trained on the task-related attribute behave on the main dataset. However, this part of the study is crucial but not 100% clear, and the links with the results of these control experiments are not fully explicit. Could the author better clarity this important point (see also Issue #1 and #3).

The reviewer raises an important point, alluding to potential differences between decoded features regarding task relevance. There are two separate sets of analyses where task relevance may have been a factor, our main analyses comparing illusion to contrast decoding, and our comparison of collinearity vs. illusion-specific processing.

In our main analysis, we are indeed reporting decoding of a task-relevant feature (illusion) and of a task-irrelevant feature (local contrast, i.e., rotation of the Pac-Man inducers). Note, however, that the Pac-Man inducers were always task-relevant, as they needed to be processed to perceive illusory triangles, so that local contrast decoding was based on task-relevant stimulus elements, even though participants did not respond to local contrast differences in the main experiment. However, we also ran control analyses testing the effect of task-relevance on local contrast decoding in our independent training data set and in another (independent) study, where local contrast was, in separate experimental blocks, task-relevant or task-irrelevant. The results are reported in the Supplementary Text and in Figure S5. In brief, task-relevance did not improve early (70–95 ms) decoding of local contrast. We are thus confident that the comparison of local contrast to illusion decoding in our main analysis was not substantially affected by differences in task relevance. In our previous manuscript version, we referred to these control analyses only in the collinearity-vs-illusion section of the Results. In our revision, we added the following in the Results section comparing illusion to contrast decoding:

“In the light of evidence showing that unconscious processing is susceptible to conscious top-down influences (Kentridge et al., 2004; Kiefer & Brendel, 2006; Naccache et al., 2002), we ran control analyses showing that early local contrast decoding was not improved by rendering contrast task-relevant (see Supplementary Information and Fig. S5), indicating that these differences between illusion and contrast decoding did not reflect differences in task-relevance.”

In addition to our main analysis, there is the concern that our comparison of collinearity vs. illusion-specific processing may have been affected by differences in task-relevance between the stimuli inducing the non-illusory triangle (the “two-legged white circles”, collinearity-only) and the stimuli inducing the Kanizsa illusion (the PacMan inducers, collinearity-plus-illusion). We would like to emphasize that in our main analysis classifiers were always used to decode T2 illusion presence vs. absence (collinearity-plus-illusion), and never to decode T2 collinearity-only. To distinguish collinearity-only from collinearity-plus-illusion processing, we only varied the training data (training classifiers on collinearity-only or collinearity-plus-illusion), using the independent training data set, where collinearity-only and collinearity-plus-illusion (and rotation) were task-relevant (in separate blocks). As discussed in the Supplementary Information, for this analysis approach to be valid, collinearity-only processing should be similar for the illusory and the non-illusory triangle, and this is what control analyses demonstrated (Fig. S7). In any case, general task-relevance was equated for the collinearity-only and the collinearity-plus-illusion classifiers.

Finally, in supplementary Figure 6 we also show that our main results reported in Figure 2 (discussed at the top of this response) were very similar when the classifiers were trained on the independent localizer dataset in which each stimulus feature could be task-relevant.

Together, for the reasons described above, we believe that differences in EEG decoding performance across these three stimulus levels did are unlikely to depend also depend on a “task-relevance” asymmetry.

Issue #2: Following on my previous point the authors should better mention the concept of conscious influences on unconscious processing that led to a full revision of the notion of automaticity in cognitive science [1 , 2 , 3 , 4]. For instance, the discovery that conscious endogenous temporal and spatial attention modulate unconscious subliminal processing paved the way to this revision. This concept raises the importance of Issue#1: equating performance on the main task across AB and masking is not enough to guarantee that differences of neural processing of the unattended attributes of T2 (i.e.: task-unrelated attributes) are not, in part, due to this asymmetry rather than to a systematic difference of unconscious processing strengtsh [5 , 6-8]. Obviously, the reported differences for real-triangle decoding between AB and masking cannot be totally explained by such a factor (because this is a task-unrelated attribute for both AB and masking conditions), but still this issue should be better introduced, addressed, clarified (Issue #1 and #3) and discussed.

We would like to refer to our response to the previous point: Control analyses for local contrast decoding showed that task relevance had no influence on our marker for feedforward processing. Most importantly, as outlined above, we did not perform real-triangle decoding – all our decoding analyses focused on comparing collinearity-only vs. collinearity-plus-illusion were run on the task-relevant T2 illusion (decoding its presence vs. absence). The key difference was solely the training set, where the collinearity-only classifier was trained on the (task-relevant) real triangle and the collinearity-plus-illusion classifier was trained on the (task-relevant) Kanizsa triangle. Thus, overall task relevance was controlled in these analyses.

In our revision, we are now also citing the studies proposed by the reviewer, when discussing the control analyses testing for an effect of task-relevance on local contrast decoding:

“In the light of evidence showing that unconscious processing is susceptible to conscious top-down influences (Kentridge et al., 2004; Kiefer & Brendel, 2006; Naccache et al., 2002), we ran control analyses showing that early local contrast decoding was not improved by rendering contrast task-relevant (see Supplementary Information and Fig. S5), indicating that these differences between illusion and contrast decoding did not reflect differences in task-relevance.”

Issue #3: In terms of clarity, I would suggest the authors to add a synthetic figure providing an overall view of all pairs of intra and cross-conditions decoding analyses and mentioning main task for training and testing sets for each analysis (see my previous and related points). Indeed, at one point, the reader can get lost and this would not only strengthen accessibility to the detailed picture of results, but also pinpoint the limits of the work (see previous point).

We understand the point the reviewer is raising and acknowledge that some of our analyses, in particular those using different training and testing sets, may be difficult to grasp. But given the variety of different analyses using different training and testing sets, different temporal windows, as well as different stimulus features, it was not possible to design an intuitive synthetic figure summarizing the key results. We hope that the added text in the Results and Discussion section will be sufficient to guide the reader through our set of analyses.

In our revision, we are now more clearly highlighting that, in addition to presenting the key results in our main text that were based on training classifiers on the T1 data, “we replicated all key findings when training the classifiers on an independent training set where individual stimuli were presented in isolation (Fig. 3A, results in the Supplementary Information and Fig. S6).” For this, we added a schematic showing the procedure of the independent training set to Figure 3, more clearly pointing the reader to the use of a separate training data set.

Issue #4: In the light of these findings the authors should discuss more thoroughly the question of unconscious high-level representations in masking versus AB: in particular, a longstanding issue relates to unconscious semantic processing of words, numbers or pictures. According to their findings, they tend to suggest that semantic processing should be more enabled in AB than in masking. However, a rich literature provided a substantial number of results (including results from the last authors Simon Van Gaal) that tend to support the notion of unconscious semantic processing in subliminal processing (see in particular: [9 , 10 , 11 , 12 , 13]). So, and as mentioned by the authors, while there is evidence for semantic processing during AB they should better discuss how they would explain unconscious semantic subliminal processing. While a possibility could be to question the unconscious attribute of several subliminal results, the same argument also holds for AB studies. Another possible track of discussion would be to differentiate AB and subliminal perception in terms of strength and durability of the corresponding unconscious representations, but not necessarily in terms of cognitive richness. Indeed, one may discuss that semantic processing of stimuli that do not need complex spatial integration (e.g.: words or digits as compared to illusory Kanisza tested here) can still be observed under subliminal conditions.

We thank the reviewer for pointing us to this shortcoming of our previous Discussion. Note that our data does not directly speak to the question of high-level unconscious representations in masking vs AB, because such conclusions would hinge on the operational definition of consciousness one adheres to (also see response to Reviewer 1). Nevertheless, we do follow the reviewer’s suggestions and added the following in the Discussion (also addressing a point about other forms of attention raised by Reviewer 1):

“Clearly, these findings do not imply that unconscious high-level (e.g., semantic) processing can only occur during inattention, nor do they necessarily generalize to other forms of inattention. Indeed, while the AB represents a prime example of late attentional filtering, other ways of inducing inattention or distraction (e.g., by manipulating spatial attention) may filter information earlier in the processing hierarchy (e.g., Luck & Hillyard, 1994 vs. Vogel et al., 1998).”

And, in a following paragraph in the Discussion:

“Such deep feedforward processing can be sufficient for unconscious high-level processing, as indicated by a rich literature demonstrating high-level (e.g., semantic) processing during masking (Kouider & Dehaene, 2007; Van den Bussche et al., 2009; van Gaal & Lamme, 2012). Thus, rather than enabling high-level unconscious processing, preserved local recurrency during inattention may afford other processing advantages linked to its proposed role in perceptual integration (Lamme, 2020), such as integration of stimulus elements over space or time.

**Reviewer #3 (Recommendations For The Authors):**
(1) The objective of Fahrenfort et al., 2017 seems very similar to that of the current study. What are the main differences between the two studies? Moreover, Fahrenfort et al., 2017 conducted similar decoding analyses to those performed in the current study.Which results were replicated in the current study, and which ones are novel? Highlighting these differences in the manuscript would be beneficial.

We now provide a more comprehensive coverage of the study by Fahrenfort et al., 2017. In the Introduction, we added a brief summary of the key findings, highlighting that this study’s findings could have reflected differences in task performance rather than differences between masking and AB:

“For example, Fahrenfort and colleagues (2017) found that illusory surfaces could be decoded from electroencephalogram (EEG) data during the AB but not during masking. This was taken as evidence that local recurrent interactions, supporting perceptual integration, were preserved during inattention but fully abolished by masking. However, masking had a much stronger behavioral effect than the AB, effectively reducing task performance to chance level. Indeed, a control experiment using weaker masking, which resulted in behavioral performance well above chance similar to the main experiment’s AB condition, revealed some evidence for preserved local recurrent interactions also during masking. However, these conditions were tested in separate experiments with small samples, precluding a direct comparison of perceptual vs. attentional blindness at matched levels of behavioral performance. To test …”

In the Results , we are now also highlighting this key advancement by directly referencing the previous study:

“Thus, whereas in previous studies task performance was considerably higher during the AB than during masking (e.g., Fahrenfort et al., 2017), in the present study the masked and the AB condition were matched in both measures of conscious access.” When reporting the EEG decoding results in the Results section, we continuously cite the Fahrenfort et al. (2017) study to highlight similarities in the study’s findings. We also added a few sentences explicitly relating the key findings of the two studies:

“This suggests that the AB allowed for greater local recurrent processing than masking, replicating the key finding by Fahrenfort and colleagues (2017). Importantly, the present result demonstrates that this effect reflects the difference between the perceptual vs. attentional manipulation rather than differences in behavior, as the masked and the AB condition were matched for perceptual performance and metacognition.”

“This similarity between behavior and EEG decoding replicates the findings of Fahrenfort and colleagues (2017) who also found a striking similarity between late Kanizsa decoding (at 406 ms) and behavioral Kanizsa detection. These results indicate that global recurrent processing at these later points in time reflected conscious access to the Kanizsa illusion.”

We also more clearly highlighted where our study goes beyond Fahrenfort et al.’s (2017), e.g., in the Results:

“The addition of this element of collinearity to our stimuli was a key difference to the study by Fahrenfort and colleagues (2017), allowing us to compare non-illusory triangle decoding to illusory triangle decoding in order to distinguish between collinearity and illusion-specific processing.”

And in the Discussion:

“Furthermore, the addition of line segments forming a non-illusory triangle to the stimulus employed in the present study allowed us to distinguish between collinearity and illusion-specific processing.”

Also, in the Discussion, we added a paragraph “summarizing which results were replicated in the current study, and which ones are novel”, as suggested by the reviewer:

“This pattern of results is consistent with a previous study that used EEG to decode Kanizsa-like illusory surfaces during masking and the AB (Fahrenfort et al., 2017). However, the present study also revealed some effects where Fahrenfort and colleagues (2017) failed to obtain statistical significance, likely reflecting the present study’s considerably larger sample size and greater statistical power. For example, in the present study the marker for feedforward processing was weakly but significantly impaired by masking, and the marker for local recurrency was significantly impaired not only by masking but also by the AB, although to a lesser extent. Most importantly, however, we replicated the key findings that local recurrent processing was more strongly impaired by masking than by the AB, and that global recurrent processing was similarly impaired by masking and the AB and closely linked to task performance, reflecting conscious access. Crucially, having matched the key conditions behaviorally, the present finding of greater local recurrency during the AB can now unequivocally be attributed to the attentional vs. perceptual manipulation of consciousness.”

Finally, we changed the title to “Distinct neural mechanisms underlying perceptual and attentional impairments of conscious access despite equal task performance” to highlight one of the crucial differences between the Fahrenfort et al., study and this study, namely the fact that we equalized task performance between the two critical conditions (AB and masking).

(2) It is not clear from the text the link between the current study and the literature on the role of lateral and feedback connections in consciousness (Lamme, 2020). A better explanation is needed.

To our knowledge, consciousness theories such as recurrent processing theory by Lamme make currently no distinction between the role of lateral and feedback connections for consciousness. The principled distinction lies between unconscious feedforward processing and phenomenally conscious or “preconscious” local recurrent processing, where local recurrency refers to both lateral (or horizontal) and feedback connections. We added a sentence in the Discussion:

“As current theories do not distinguish between the roles of lateral vs. feedback connections for consciousness, the present findings may enrich empirical and theoretical work on perceptual vs. attentional mechanisms of consciousness …”

(3) When training on T1 and testing on T2, EEG data showed an early peak in local contrast classification at 75-95 ms over posterior electrodes. The authors stated that this modulation was only marginally affected by masking (and not at all by AB); however, the main effect of masking is significant. Why was this effect interpreted as nonrelevant?

Following this and Reviewer 1’s comment, we changed the wording from “marginal” to “weak but significant.” We considered this effect “weak” and of lesser relevance, because its Bayes factor indicated that the alternative hypothesis was only 1.31 times more likely than the null hypothesis of no effect, representing only “anecdotal” evidence, which is in sharp contrast to the robust effects of the consciousness manipulations on illusion decoding reported later. Furthermore, later ANOVAs comparing the effect of masking on contrast vs. illusion decoding revealed much stronger effects on illusion decoding than on contrast decoding (BFs>3.59×10^4^).

(4) The decoding analysis on the illusory percept yielded two separate peaks of decoding, one from 200 to 250 ms and another from 275 to 475 ms. The early component was localized occipitally and interpreted as local sensory processing, while the late peak was described as a marker for global recurrent processing. This latter peak was localized in the parietal cortex and associated with the P300. Can the authors show the topography of the P300 evoked response obtained from the current study as a comparison? Moreover, source reconstruction analysis would probably provide a better understanding of the cortical localization of the two peaks.

Figure S4 now shows the P300 from electrode Pz, demonstrating a stronger positivity between 375 and 475 ms when the illusory triangle was present than when it was absent. We did not run a source reconstruction analysis.

(5) The authors mention that the behavioural results closely resembled the pattern of the second decoding peak results. However, they did not show any evidence for this relationship. For instance, is there a correlation between the two measures across or within participants? Does this relationship differ between the illusion report and the confidence rating?

This relationship became evident from simply eyeballing the results figures: Both in behavior and EEG decoding performance dropped from the both-manipulations condition to the AB and masked conditions, while these conditions did not differ significantly. Following a similar observation of a close similarity between behavior and the second/late illusion decoding peak in the study by Fahrenfort et al. (2017), we adopted their analysis approach and ran two additional ANOVAs, adding “measure” (behavior vs. EEG) as a factor. For this analysis, we dropped the both-manipulations condition due to scale restrictions (as noted in footnote 1: “We excluded the bothmanipulations condition from this analysis due to scale restrictions: in this condition, EEG decoding at the second peak was at chance, while behavioral performance was above chance, leaving more room for behavior to drop from the masked and AB condition.”). The analysis revealed that there were no interactions with condition:

“The pattern of behavioral results, both for perceptual performance and metacognitive sensitivity, closely resembled the second decoding peak: sensitivity in all three metrics dropped from the no-manipulations condition to the masked and AB conditions, while sensitivity did not differ significantly between these performancematched conditions (Fig. 2C). Two additional rm ANOVAs with the factors measure (behavior, second EEG decoding peak) and condition (no-manipulations, masked, AB)^*1*^ for perceptual performance and metacognitive sensitivity revealed no significant interaction performance: F2,58=0.27, P=0.762, BF01=8.47; metacognition: F2,58 who also found a striking similarity between late Kanizsa decoding (at 406 ms) and behavioral Kanizsa detection. These results indicate that global recurrent processing at these later points in time reflected conscious access to the Kanizsa illusion.”

(6) The marker for illusion-specific processing emerged later (200-250 ms), with the nomanipulation decoding performing better after training on the illusion than the nonillusory triangle. This difference emerged only in the AB condition, and it was fully abolished by masking. The authors confirmed that the illusion-specific processing was not affected by the AB manipulations by running a rm ANOVA which did not result in a significant interaction between condition and training set. However, unlike the other non-significant results, a Bayes Factor is missing here.

We added Bayes factors to all (significant and non-significant) rm ANOVAs.

(7) The same analysis yielded a second illusion decoding peak at 375-475 ms. This effect was impaired by both masking and AB, with no significant differences between the two conditions. The authors stated that this result was directly linked to behavioural performance. However, it is not clear to me what they mean (see point 5).

We added analyses comparing behavior and EEG decoding directly (see our response to point 5).

(8) The introduction starts by stating that perceptual and attentional processes differently affect consciousness access. This differentiation has been studied thoroughly in the consciousness literature, with a focus on how attention differs from consciousness (e.g., Koch & Tsuchiya, TiCS, 2007; Pitts, Lutsyshyna & Hillyard, Phil. Trans. Roy. Soc. B Biol. Sci., 2018). The authors stated that "these findings confirm and enrich empirical and theoretical work on perceptual vs. attentional mechanisms of consciousness clearly distinguishing and specifying the neural profiles of each processing stage of the influential four-stage model of conscious experience". I found it surprising that this aspect was not discussed further. What was the state of the art before this study was conducted? What are the mentioned neural profiles? How did the current results enrich the literature on this topic?

We would like to point out that our study is not primarily concerned with the conceptual distinction between consciousness and attention, which has been the central focus of e.g., Koch and Tsuchiuya (2007). While this literature was concerned with ways to dissociate consciousness and attention, we tacitly assumed that attention and consciousness are now generally considered as different constructs. Our study is thus not dealing with dissociations between attention and consciousness, nor with the distinction between phenomenal consciousness and conscious access, but is concerned with different ways of impairing conscious access (defined as the ability to report about a stimulus), either via perceptual or via attentional manipulations. For the state of the art before the study was conducted, we would like to refer to the motivation of our study in the Introduction, e.g., previous studies’ difficulties in unequivocally linking greater local recurrency during attentional than perceptual blindness to the consciousness manipulation, given performance confounds (we expanded this Introduction section). We also expanded a paragraph in the discussion to remind the reader of the neural profiles of the 4-stage model and to highlight the novelty of our findings related to the distinction between lateral and feedback processes:

“As current theories do not distinguish between the roles of lateral vs. feedback connections for consciousness, the present findings may enrich empirical and theoretical work on perceptual vs. attentional mechanisms of consciousness (Block, 2005; Dehaene et al., 2006; Hatamimajoumerd et al., 2022; Lamme, 2010; Pitts et al., 2018; Sergent & Dehaene, 2004), clearly distinguishing the neural profiles of each processing stage of the influential four-stage model of conscious experience (Fig. 1A). Along with the distinct temporal and spatial EEG decoding patterns associated with lateral and feedback processing, our findings suggest a processing sequence from feedforward processing to local recurrent interactions encompassing lateral-tofeedback connections, ultimately leading to global recurrency and conscious report.”

(9) When stating that this is the first study in which behavioural measures of conscious perception were matched between the attentional blink and masking, it would be beneficial to highlight the main differences between the current study and the one from Fahrenfort et al., 2017, with which the current study shares many similarities in the experimental design (see point 1).

We would like to refer the reviewer to our response to point (1), where we detail how we expanded the discussion of similarities and differences between our present study and Fahrenfort et al. (2017).

(10) The discussion emphasizes how the current study "suggests a processing sequence from feedforward processing to local recurrent interactions encompassing lateral-to-feedback connections, ultimately leading to global recurrency and conscious report". For transparency, it is though important to highlight that one limit of the current study is that it does not provide direct evidence for the specified types of connections (see point 6).

We added a qualification in the Discussion section:

“Although the present EEG decoding measures cannot provide direct evidence for feedback vs. lateral processes, based on neurophysiological evidence, …”

Furthermore, we added this qualification in the Discussion section:

“It should be noted that the not all neurophysiological evidence unequivocally links processing of collinearity and of the Kanizsa illusion to lateral and feedback processing, respectively (Angelucci et al., 2002; Bair et al., 2003; Chen et al., 2014), so that overlap in decoding the illusory and non-illusory triangle may reflect other mechanisms, for example feedback processing as well.”

References

Angelucci, A., Levitt, J. B., Walton, E. J. S., Hupe, J.-M., Bullier, J., & Lund, J. S. (2002). Circuits for local and global signal integration in primary visual cortex. The Journal of Neuroscience: The Official Journal of the Society for Neuroscience, 22(19), 8633–8646.

Bair, W., Cavanaugh, J. R., & Movshon, J. A. (2003). Time course and time-distance relationships for surround suppression in macaque V1 neurons. The Journal of Neuroscience: The Official Journal of the Society for Neuroscience, 23(20), 7690–7701.

Block, N. (2005). Two neural correlates of consciousness. Trends in Cognitive Sciences, 9(2), 46–52.

Chen, M., Yan, Y., Gong, X., Gilbert, C. D., Liang, H., & Li, W. (2014). Incremental integration of global contours through interplay between visual cortical areas. Neuron, 82(3), 682–694.

Dehaene, S., Changeux, J.-P., Naccache, L., Sackur, J., & Sergent, C. (2006). Conscious, preconscious, and subliminal processing: a testable taxonomy. Trends in Cognitive Sciences, 10(5), 204–211.

Hatamimajoumerd, E., Ratan Murty, N. A., Pitts, M., & Cohen, M. A. (2022). Decoding perceptual awareness across the brain with a no-report fMRI masking paradigm. Current Biology: CB. https://doi.org/10.1016/j.cub.2022.07.068

JASP Team. (2024). JASP (Version 0.19.0)[Computer software]. https://jasp-stats.org/ Kentridge, R. W., Heywood, C. A., & Weiskrantz, L. (2004). Spatial attention speeds discrimination without awareness in blindsight. Neuropsychologia, 42(6), 831– 835.

Kiefer, M., & Brendel, D. (2006). Attentional Modulation of Unconscious “Automatic” Processes: Evidence from Event-related Potentials in a Masked Priming Paradigm. Journal of Cognitive Neuroscience, 18(2), 184–198.

Kouider, S., & Dehaene, S. (2007). Levels of processing during non-conscious perception: a critical review of visual masking. Philosophical Transactions of the Royal Society B: Biological Sciences, 362(1481), 857–875.

Lamme, V. A. F. (2010). How neuroscience will change our view on consciousness. Cognitive Neuroscience, 1(3), 204–220.

Luck, S. J., & Hillyard, S. A. (1994). Electrophysiological correlates of feature analysis during visual search. Psychophysiology, 31(3), 291–308.

Naccache, L., Blandin, E., & Dehaene, S. (2002). Unconscious masked priming depends on temporal attention. Psychological Science, 13(5), 416–424.

Pitts, M. A., Lutsyshyna, L. A., & Hillyard, S. A. (2018). The relationship between attention and consciousness: an expanded taxonomy and implications for ‘noreport’ paradigms. Philosophical Transactions of the Royal Society of London. Series B, Biological Sciences, 373(1755), 20170348.

Sergent, C., & Dehaene, S. (2004). Is consciousness a gradual phenomenon? Evidence for an all-or-none bifurcation during the attentional blink. Psychological Science, 15(11), 720–728.

Van den Bussche, E., Van den Noortgate, W., & Reynvoet, B. (2009). Mechanisms of masked priming: a meta-analysis. Psychological Bulletin, 135(3), 452–477. van Gaal, S., & Lamme, V. A. F. (2012). Unconscious high-level information processing: implication for neurobiological theories of consciousness: Implication for neurobiological theories of consciousness. The Neuroscientist: A Review Journal Bringing Neurobiology, Neurology and Psychiatry, 18(3), 287–301.

Vogel, E. K., Luck, S. J., & Shapiro, K. L. (1998). Electrophysiological evidence for a postperceptual locus of suppression during the attentional blink. Journal of Experimental Psychology. Human Perception and Performance, 24(6), 1656– 1674.